# AP-4-mediated axonal transport controls endocannabinoid production in neurons

Alexandra K. Davies [1✉], Julian E. Alecu [2], Marvin Ziegler [2,3], Catherine G. Vasilopoulou [1], Fabrizio Merciai[1,4], Hellen Jumo[2], Wardiya Afshar-Saber [2], Mustafa Sahin [2,5], Darius Ebrahimi-Fakhari [2,6] & Georg H. H. Borner [1,6✉]

The adaptor protein complex AP-4 mediates anterograde axonal transport and is essential for axon health. AP-4-deficient patients suffer from a severe neurodevelopmental and neuro-degenerative disorder. Here we identify DAGLB (diacylglycerol lipase-beta), a key enzyme for generation of the endocannabinoid 2-AG (2-arachidonoylglycerol), as a cargo of AP-4 vesicles. During normal development, DAGLB is targeted to the axon, where 2-AG signalling drives axonal growth. We show that DAGLB accumulates at the trans-Golgi network of AP-4-deficient cells, that axonal DAGLB levels are reduced in neurons from a patient with AP-4 deficiency, and that 2-AG levels are reduced in the brains of AP-4 knockout mice. Importantly, we demonstrate that neurite growth defects of AP-4-deficient neurons are rescued by inhibition of MGLL (monoacylglycerol lipase), the enzyme responsible for 2-AG hydrolysis. Our study supports a new model for AP-4 deficiency syndrome in which axon growth defects arise through spatial dysregulation of endocannabinoid signalling.

[1] Department of Proteomics and Signal Transduction, Max Planck Institute of Biochemistry, Martinsried 82152, Germany. [2] Department of Neurology, The F.M. Kirby Neurobiology Center, Boston Children's Hospital, Harvard Medical School, Boston, MA 02115, USA. [3] Department of Functional Neuroanatomy, Institute of Anatomy and Cell Biology, Heidelberg University, INF 307, Heidelberg 69120, Germany. [4] Department of Pharmacy and PhD Program in Drug Discovery and Development, University of Salerno, 84084 Fisciano, SA, Italy. [5] Rosamund Stone Zander Translational Neuroscience Center, Boston Children's Hospital, Harvard Medical School, Boston, MA 02115, USA. [6] These authors jointly supervised this work: Darius Ebrahimi-Fakhari, Georg H. H. Borner. ✉email: davies@biochem.mpg.de; borner@biochem.mpg.de

Axon development and maintenance critically rely on tightly regulated anterograde transport. The factors that promote axonal growth and the machinery that removes dysfunctional components must efficiently reach the distal axon. Accordingly, mutations in the axonal transport machinery are implicated in a broad range of neurodevelopmental and neurodegenerative diseases[1]. We and others recently discovered that adaptor protein complex 4 (AP-4) is required for the axonal delivery of vesicles containing the autophagy protein ATG9A[2–5], a lipid scramblase that promotes the expansion of autophagosomal membrane[6,7]. Biallelic loss-of-function mutations in any of the four subunits of AP-4 (encoded by the genes *AP4B1*, *AP4E1*, *AP4M1*, *AP4S1*) cause AP-4 deficiency syndrome, a severe disease characterised by global developmental delay, intellectual disability, seizures and progressive spasticity, with the onset of symptoms in early infancy[8–10]. There is currently no treatment, and a steep rise in diagnoses over the last three years[11] is fuelling efforts to understand the mechanisms that cause neuronal pathology in AP-4 deficiency.

Neuron-specific knockout of the core autophagy genes *Atg5* or *Atg7* abolishes neuronal autophagy and leads to neurodegeneration[12,13]. Since ATG9A is also a core component of the autophagy machinery, the current model for AP-4 deficiency suggests that lack of ATG9A in the distal axon leads to impaired autophagosome biogenesis, causing axonal degeneration. However, defects in neuronal autophagy are insufficient to explain all aspects of the disease. In vitro, neurons from embryonic *Ap4e1* knockout mice have axonal growth defects including reduced length and branching[5], and neurite outgrowth is also impaired in *Atg9a* knockout neurons[14]. In contrast, axon length is normal in cultured neurons from autophagy-deficient *Atg7* knockout mouse embryos[14]. This suggests that AP-4-derived vesicles have unidentified roles in neurodevelopment that are independent of their role in autophagy.

Unbiased proteomic approaches have yielded a consensus set of AP-4 vesicle cargoes and accessory proteins[2,15,16] (summarised in Supplementary Table 1). Our spatial proteomics method, Dynamic Organellar Maps, provided a powerful screen for proteins with an altered distribution in AP-4-deficient cells, leading to the identification of the AP-4 cargo proteins ATG9A, SERINC1, and SERINC3[2]. Using this approach, we here identify DAGLB (diacylglycerol lipase-beta) as a cargo protein of AP-4 vesicles. DAGLB is a key enzyme for the generation of 2-arachidonoylglycerol (2-AG), the most abundant endocannabinoid in the brain. In developing neurons, an unknown pathway targets DAGLB to the distal axon, where 2-AG is required to promote axonal growth and guidance[17,18]. Our data strongly support that AP-4 mediates anterograde axonal transport of DAGLB. We therefore suggest that AP-4 provides a missing link in the regulation of endocannabinoid signalling, and that neurodevelopmental aspects of AP-4 deficiency syndrome may arise through disruption of the spatial control of 2-AG synthesis.

## Results and discussion
### Spatial proteomics reveals DAGLB as an AP-4 cargo protein.
We previously used our spatial proteomics method, Dynamic Organellar Maps, to identify cargoes of the AP-4 vesicle pathway[2]. In this approach, a map of the cell is generated by combining cell fractionation with mass spectrometry-based quantification to provide proteome-wide protein localisation information[19]. We compared maps from wild-type, *AP4B1* knockout and *AP4E1* knockout HeLa cells in biological duplicate, to identify proteins with an altered subcellular localisation in the AP-4-deficient cells. Using very stringent data filters (false discovery rate (FDR) < 1%), we identified three proteins with a localisation shift (Supplementary Fig. 1a)—ATG9A, SERINC1 and SERINC3—which we then validated as bona fide AP-4 cargoes by imaging[2].

To investigate if AP-4 may have additional cargo proteins, we now performed a more sensitive exploratory analysis of our data (Fig. 1a and Supplementary Fig. 1b; see "Methods" for details). We identified eight additional hits, with an estimated FDR of approximately 25% (inset plot of Fig. 1a). Since some false positives were expected, we applied several further filters to pinpoint the most relevant shifts. First, we visualised the localisation of our new hits on an organellar map of wild-type HeLa cells[19] (Fig. 1b). Three proteins, DAGLB, PTPN9 and LNPEP, mapped to the endosomal cluster, like the known AP-4 cargo proteins. Of these, only DAGLB displayed a shift profile that was highly correlated with those of ATG9A, SERINC1 and SERINC3 (Fig. 1c, d), which is strongly indicative of an AP-4 cargo. In contrast, PTPN9 and LNPEP are unlikely to be AP-4 vesicle proteins because their shift profiles do not correlate with those of AP-4 cargoes (Fig. 1c and Supplementary Fig. 1c, d). Nevertheless, PTPN9 and LNPEP have highly similar shift profiles (Supplementary Fig. 1c, d), suggesting they may be mistrafficked together, perhaps as a secondary effect of AP-4 deficiency.

In our previous study, we found that ATG9A, SERINC1 and SERINC3 were co-immunoprecipitated with the AP-4 complex[2]. We reanalysed this proteomic dataset and found that DAGLB was indeed co-enriched, along with known AP-4 vesicle proteins, while PTPN9 and LNPEP were absent (Fig. 1e). Combined with our mapping data, this strongly supports that DAGLB is an AP-4 cargo protein.

DAGLB is a serine lipase that hydrolyses diacylglycerol (DAG) to generate the signalling lipid 2-AG[17]. 2-AG is an agonist for the $CB_1$ and $CB_2$ cannabinoid receptors ($CB_1R/CB_2R$), which have important functions in axonal growth and guidance during development[20]. As axonal growth defects arise through an unknown mechanism in AP-4-deficient neurons, we investigated AP-4-dependent localisation of DAGLB in detail.

### DAGLB accumulates at the TGN of AP-4 deficient cells.
In the absence of AP-4, ATG9A accumulates at the trans-Golgi network (TGN) of diverse cell types, including neuronal and non-neuronal cells[2–5,21]. To assess whether DAGLB exhibits a similar mis-sorting phenotype in AP-4-deficient cells, we used immunofluorescence microscopy. In wild-type HeLa cells DAGLB was observed in fine puncta throughout the cell, with increased density in the region of the TGN (Fig. 2a). This distribution is strikingly similar to that of ATG9A[2,22]. In *AP4B1* knockout cells there was a significant increase in the ratio of DAGLB intensity between the TGN and the rest of the cell, which was rescued by stable expression of the missing AP4B1 subunit (Fig. 2a, b). A similar increase in the amount of DAGLB at the TGN was observed in cells treated with siRNA to knock down AP-4 (Supplementary Fig. 2a, b). These data suggest that AP-4 deficiency causes a partial block of DAGLB export from the TGN. The effect was not as strong as that observed for ATG9A[2], but this is consistent with the lower movement score of DAGLB in our Dynamic Organellar Maps analysis (Fig. 1a). Next, we examined DAGLB distribution after depletion of AP-4 in neuroblastoma-derived SH-SY5Y cells[2] that had been neuronally differentiated by the addition of retinoic acid and BDNF. The AP-4 depleted SH-SY5Y cells showed a higher ratio of DAGLB at the TGN than control cells (Fig. 2c, d). This increase was more pronounced than in HeLa cells, indicating that AP-4 dependent sorting of DAGLB may be particularly important in polarised cells. The ratio of DAGLB at the TGN was also

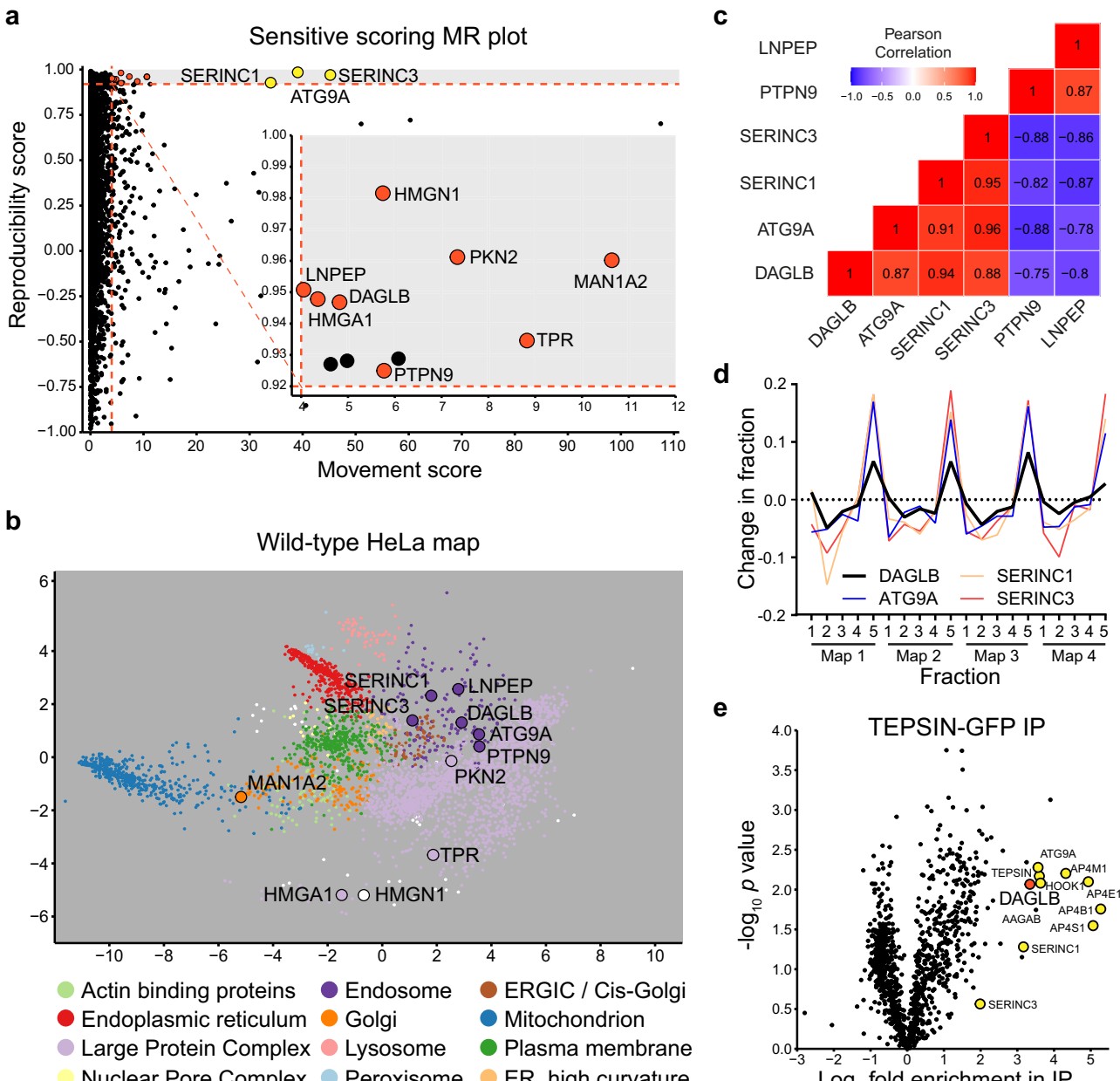

**Fig. 1 Sensitive analysis of Dynamic Organellar Maps identifies DAGLB as an AP-4 cargo protein. a** Dynamic Organellar Maps of *AP4B1* knockout (KO) and *AP4E1* KO HeLa cells were compared to maps of wild-type HeLa cells (each in duplicate, totalling four comparisons). Sensitive statistical scoring was used to detect proteins with a significantly altered distribution in the AP-4 KO cells that were not detected in a previous stringent analysis[2] (see Methods for details). For 3926 proteins profiled across all maps, the 'MR' plot displays the median magnitude of shift (M) and the mean within-clone reproducibility of shift direction (R). The known AP-4 cargo proteins, ATG9A, SERINC1 and SERINC3 (marked in yellow), were identified with high M and R scores, as expected. The inset plot displays 8 additional hits (marked in red) whose subcellular localisation was significantly and reproducibly shifted in the AP-4 KO lines, with a false discovery rate of ~25%. Proteins that passed the M and R cut-offs but had poor across-clone reproducibility were not considered as hits (marked in black). **b** The hits from (**a**) are highlighted on a principal component analysis (PCA)-based visualisation of a deep Dynamic Organellar Map of wild-type HeLa cells[19] (combined data from six replicate maps [http://mapofthecell.biochem.mpg.de/]). Each scatter point represents a protein and proximity indicates similar fractionation profiles. Colours indicate subcellular compartment assignment by support vector machine-based classification (white indicates compartment unassigned). Three hits, DAGLB, PTPN9 and LNPEP, map to the endosomal cluster (dark purple), like known AP-4 cargo proteins. **c** Heat map showing pairwise Pearson correlations between the shift profiles (abundance distribution profiles from knockout maps subtracted from the profiles from control maps) of DAGLB, ATG9A, SERINC1, SERINC3, PTPN9 and LNPEP. **d** The shift profiles of DAGLB, ATG9A, SERINC1 and SERINC3 are highly correlated, making DAGLB a strong new candidate AP-4 vesicle protein. Maps 1 and 2 are *AP4B1* KO maps subtracted from wild-type maps; Maps 3 and 4 are *AP4E1* KO maps subtracted from wild-type maps. Fraction numbers 1-5 refer to increasing centrifugation speeds. **e** High-sensitivity low-detergent immunoprecipitations (IP) from HeLa cells stably expressing the AP-4 associated protein TEPSIN-GFP were analysed by SILAC-based quantitative mass spectrometry[2]. Data were analysed in comparison to mock immunoprecipitations from wild-type HeLa cells with a two-tailed one sample ratio *t* test against 0 (each in triplicate, *n* = 3). DAGLB (marked in red) was highly enriched along with known AP-4 vesicle proteins (marked in yellow). Source data are provided as a Source Data file.

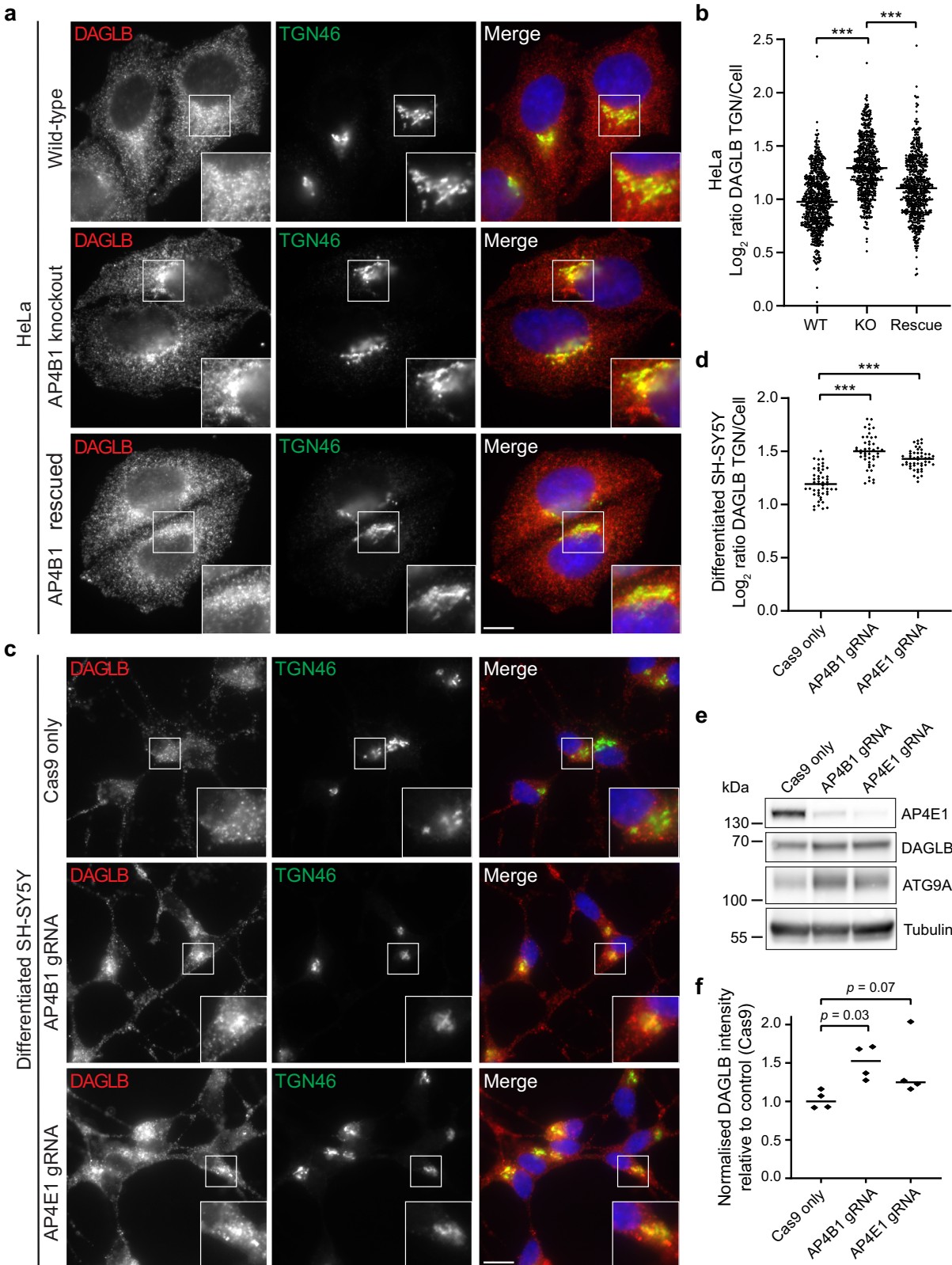

increased in AP-4-depleted SH-SY5Y cells in their undifferentiated state (Supplementary Fig. 2c, d). Collectively, these observations confirm DAGLB as an AP-4 cargo in HeLa and SH-SY5Y cells, and suggest that TGN export of DAGLB is a ubiquitous function of the AP-4 pathway. Further supporting this notion, RNA-based expression analysis suggests that DAGLB and AP-4 are both ubiquitously expressed in human tissues (Supplementary Fig. 3; Human Protein Atlas[23] [http://www.proteinatlas.org]).

AP-4-deficiency increases the expression level of ATG9A in neurons and patient fibroblasts, and this serves as a robust diagnostic disease marker[4,5,21,24]. Likewise, we observed increased levels of DAGLB in whole-cell lysates from AP-4-depleted neuronally differentiated SH-SY5Y cells (Fig. 2e, f). As has been

**Fig. 2 DAGLB accumulates at the trans-Golgi network (TGN) in AP-4 knockout (KO) HeLa and neuronally differentiated SH-SY5Y cells. a** Widefield imaging of immunofluorescence double labelling of DAGLB (red) and TGN46 (green) in wild-type (WT), *AP4B1* KO, and *AP4B1* KO HeLa cells stably expressing AP4B1 (functional rescue). In the merged image, DAPI labelling of the nucleus is also shown (blue). Scale bar: 10 µm. **b** Quantification of the ratio of DAGLB labelling intensity between the TGN and the rest of the cell, in the cells shown in (**a**). The experiment was performed in biological triplicate and the graph shows combined replicate data: each datapoint indicates the $\log_2$ ratio for an individual cell (horizontal bar indicates median; $n = 655$ cells for WT; $n = 588$ cells for KO; $n = 586$ cells for Rescue; examined over three independent experiments). Data were subjected to a Kruskal–Wallis test with Dunn's Multiple Comparison Post-Test for significance: ***$p \leq 0.001$ (KO vs WT: $p = 6.4 \times 10^{-80}$; Rescue vs KO $p = 1.0 \times 10^{-27}$. **c** CRISPR-Cas9 was used to deplete AP4B1 or AP4E1 in mixed populations of SH-SY5Y cells[2]. Widefield imaging of immunofluorescence double labelling of DAGLB (red) and TGN46 (green) in control (parental Cas9-expressing), AP4B1-depleted and AP4E1-depleted neuronally differentiated SH-SY5Y cells. In the merged image, DAPI labelling of the nucleus is also shown (blue). Scale bar: 10 µm. **d** Quantification of the ratio of DAGLB labelling intensity between the TGN and the rest of the cell, in the cells shown in (**c**). The experiment was performed in biological triplicate and the graph shows combined replicate data: each datapoint indicates the $\log_2$ ratio calculated from a single image (horizontal bar indicates median; $n = 50$ images for Cas9 only; $n = 47$ images for AP4B1 gRNA; $n = 50$ images for AP4E1 gRNA; examined over three independent experiments). Data were subjected to a Kruskal-Wallis test with Dunn's Multiple Comparison Test for significance: ***$p \leq 0.001$ (Cas9 vs AP4B1: $p = 1.1 \times 10^{-15}$; Cas9 vs AP4E1: $p = 1.3 \times 10^{-9}$). **e** Western blots of whole-cell lysates from the cells shown in (**c**); alpha-tubulin serves as a loading control. The levels of DAGLB were increased in AP-4-depleted cells, similarly to ATG9A. **f** Quantification of DAGLB from (**e**) and replicate blots, normalised to alpha-tubulin, relative to the median control (Cas9 only) level (horizontal bars indicate median). Data are from $n = 4$ blots per cell line, and three separate differentiations. Log-transformed normalised intensity values were subjected to a one-way ANOVA with Dunnett's Multiple Comparison Test for comparisons to the control. Source data are provided as a Source Data file.

suggested for ATG9A, this could represent an adaptive mechanism to compensate for the lack of directional transport of DAGLB. Alternatively, it is possible that the altered trafficking itinerary of ATG9A and DAGLB results in reduced turnover of the proteins. Either way, the fact that DAGLB behaves similarly to ATG9A in response to the lack of AP-4 further supports that DAGLB is an AP-4 cargo protein.

**DAGLB colocalizes with AP-4 vesicle proteins.** To test if DAGLB is in AP-4-derived vesicles, we used an AP-4 cargo redistribution assay, based on overexpression of the AP-4 accessory protein RUSC2. RUSC2 mediates microtubule plus-end-directed transport of AP-4 vesicles from the TGN[2], by recruiting the kinesin-1 molecular motor[25]. Overexpression of RUSC2 dramatically alters the localisation of the AP-4 cargo proteins ATG9A, SERINC1 and SERINC3 to the periphery of the cell, and this only occurs in cells that express AP-4[2]. Consistent with our hypothesis that DAGLB is also a cargo of the AP-4 pathway, it colocalised with RUSC2 in bright puncta at the periphery of HeLa cells that stably overexpress RUSC2-GFP (Fig. 3). Importantly, this was completely dependent on AP-4, as in *AP4B1* knockout cells neither DAGLB nor RUSC2 accumulated at the cell periphery, and they also no longer colocalised with each other. Transient expression of AP4B1 in the *AP4B1* knockout rescued both the peripheral localisation and the colocalisation between DAGLB and RUSC2. These observations demonstrate that DAGLB is present in AP-4 vesicles, which bud from the TGN and also contain ATG9A, SERINC1 and SERINC3[2].

Mammals have two DAG lipases, DAGLA and DAGLB[17]. However, we did not detect the expression of DAGLA by sensitive mass spectrometry analysis of HeLa whole cell proteomes[2,19]. To determine whether DAGLA is also a cargo of AP-4 vesicles, we expressed HA-tagged DAGLA or DAGLB[26] in wild-type HeLa and HeLa cells overexpressing GFP-tagged RUSC2 (Fig. 4a, b). As expected, HA-DAGLB overlapped with endogenous ATG9A in wild-type cells and relocalised to peripheral puncta in the RUSC2 overexpressing cell line (Fig. 4a). In contrast, HA-DAGLA localised to the cell surface and to reticular structures in wild-type cells, as previously reported[26]. Crucially, RUSC2 overexpression did not alter this pattern (Fig. 4b) and HA-DAGLA was not detected in the peripheral RUSC2- and ATG9A-positive puncta. These experiments suggest that, unlike DAGLB, DAGLA is not an AP-4 vesicle cargo.

To visualise colocalisation between AP-4 vesicle cargoes, we used super-resolution structured illumination microscopy (SR-

SIM). To calibrate our imaging pipeline, we first quantified colocalisation between the known AP-4 cargo proteins SERINC1, SERINC3 and ATG9A. As expected, there was considerable overlap between SERINC1/3 and ATG9A in small peripheral puncta, and colocalisation was significantly reduced by AP-4 knockdown (Fig. 5a, b and Supplementary Fig. 4a, b), consistent with our previous analysis[2]. We could not image DAGLB and ATG9A in the same cells due to antibody incompatibility, but we were able to analyse colocalisation between DAGLB and SERINC1 (Fig. 5c, d), and between DAGLB and SERINC3 (Supplementary Fig. 4c, d). The results were almost identical to those observed for ATG9A: DAGLB colocalised with the SERINCs in small peripheral puncta, and AP-4 knockdown resulted in the loss of these puncta. Taken together, our RUSC2 overexpression and SIM colocalisation assays demonstrate that DAGLB is a cargo of AP-4 vesicles.

**DAGLB is mislocalised in iPSC neurons from AP-4 patients.** We recently developed a human neuron model of AP-4 deficiency syndrome, which is based on the differentiation of iPSCs derived from patients with AP4B1 deficiency (SPG47) into excitatory cortical neurons[21]. As in primary neurons from *Ap4e1* knockout mice[4,5], ATG9A accumulates at the TGN of the AP-4-deficient human neurons and is elevated at the whole-cell level[21]. To test if DAGLB shows similar alterations, we compared the distribution of DAGLB in iPSC-derived neurons from a patient with SPG47 and their unaffected heterozygous parent (Fig. 6a, b). Indeed, DAGLB accumulated at the TGN of neurons from the AP-4-deficient patient, in a similar manner to ATG9A. To assess if this phenotype is amenable to high-throughput screening, we used our previously established automated high-throughput imaging pipeline[21] to determine the distribution of DAGLB in three independent differentiations of neurons from two SPG47 patients and their matched controls. In both patient lines, there was a reproducible increase in the area of high-intensity juxtanuclear DAGLB, which overlapped with the TGN (Fig. 6c and Supplementary Fig. 5a, b). Furthermore, the levels of DAGLB were increased in whole-cell lysates from iPSC-derived neurons from three patients with SPG47 (Fig. 6d and Supplementary Fig. 5c), as previously observed for ATG9A[21]. Finally, we quantified the number of DAGLB-positive puncta in the axons of SPG47 and control neurons and observed a significant reduction in puncta per neurite area in the AP-4-deficient neurons (Fig. 6e, f), similar to that observed for ATG9A in primary neurons of *Ap4e1* knockout mice[5]. Collectively, these data provide strong evidence

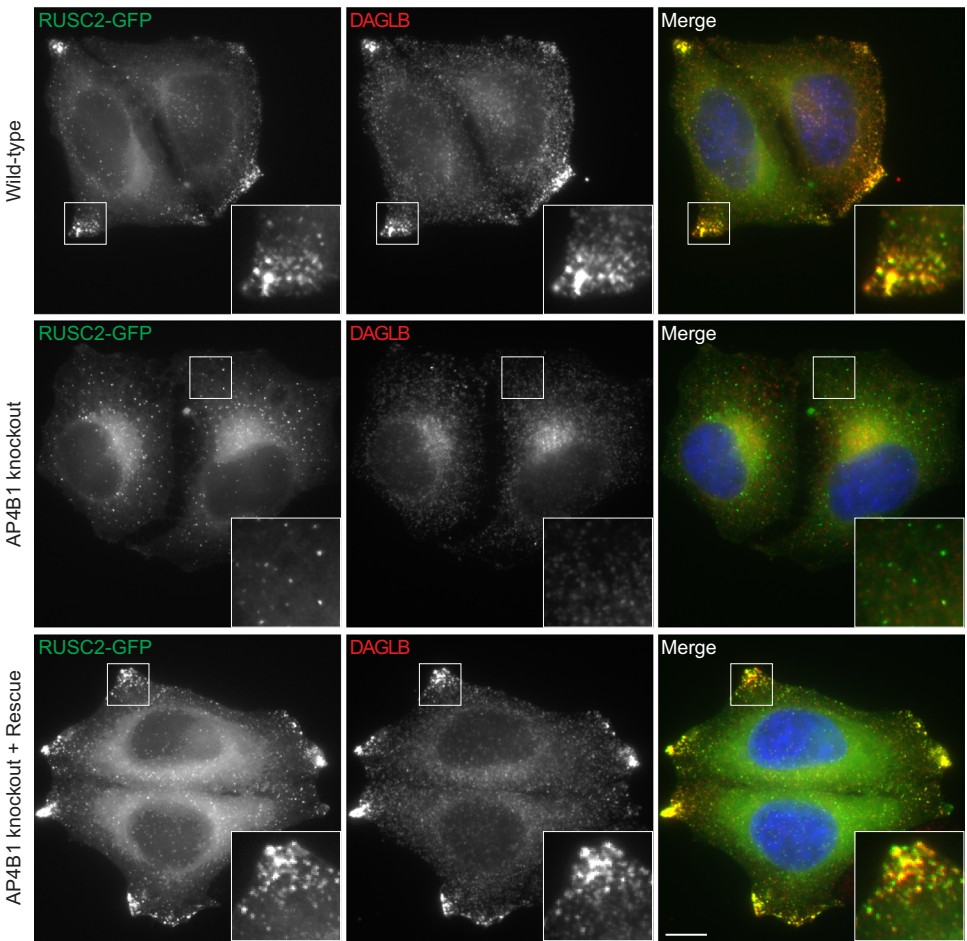

**Fig. 3 Overexpression of the AP-4 vesicle transport adaptor RUSC2 drives DAGLB to the cell periphery.** Widefield imaging of HeLa cells stably expressing RUSC2-GFP (green), labelled with anti-DAGLB (red). Top panel, wild-type; middle panel, *AP4B1* knockout; lower panel, *AP4B1* knockout with transient expression of AP4B1 (rescue). In the merged image, DAPI labelling of the nucleus is also shown (blue). The insets show accumulation of RUSC2-GFP-positive and DAGLB-positive puncta at the cell periphery, and this only occurred in the presence of AP-4 (wild-type and rescue). Images are representative of at least 20 images per condition, including three independent rescue transfections. Immunofluorescence and microscopy were performed independently from two technical replicates of separate coverslips from the same batch of cells. Consistent results were observed for cells expressing RUSC2 with an N-terminal GFP tag and with *AP4E1* knockout (data available at https://zenodo.org/record/5696988). Scale bar: 10 µm.

that AP-4 mediates axonal transport of DAGLB and that DAGLB missorting occurs in neurons of patients with AP-4 deficiency syndrome.

**2-AG levels are reduced in the brains of AP-4 knockout mice.** DAGLB is a key enzyme for generation of the endocannabinoid 2-AG, which is an important signalling lipid in the central nervous system. The major pathway for 2-AG biosynthesis is hydrolysis of diacylglycerol (DAG) at the *sn1* position by DAG lipase – DAGLA or DAGLB[17] (Fig. 7a). DAGLs are required for diverse aspects of brain function, including axonal growth during development, neurogenesis in the adult brain and retrograde synaptic signalling[20]. These functions require not only temporal control of DAGL expression, but also tight regulation of DAGL subcellular localisation. Downstream of DAGL activity, 2-AG is hydrolysed by monoacylglycerol lipase (MGLL, also known as MAGL) to produce arachidonic acid (AA)[27] (Fig. 7a). Thus, DAGL activity controls the levels of both 2-AG and AA, and both lipids decrease in parallel in the brains of *Dagla* or *Daglb* knockout mice[28]. Based on this model of the 2-AG pathway, we hypothesised that the missorting of DAGLB in AP-4 deficiency may lead to impaired DAGL activity, and hence to reduced levels of 2-AG and AA in the brain. To test this hypothesis, we used

mass spectrometry-based lipidomics to compare the levels of 2-AG and AA in brains of wild-type and *Ap4e1* knockout mice. This analysis revealed an approximately 30% reduction in 2-AG and 20% reduction in AA in the AP-4-deficient mice (Fig. 7b, c and Supplementary Fig. 6a), but no significant difference in the level of the major substrate of DAGLB, 1-stearoyl-2-arachidonoyl-sn-glycerol (SAG)[29] (Fig. 7d). Thus, our data support that DAGLB missorting in AP-4-deficient neurons leads to impaired DAGL activity and dysregulated endocannabinoid signalling.

**MGLL inhibition rescues neurite growth in patient neurons.** In developing neurons, the DAGLs are present in growing axons[17,30], which correlates with a requirement for locally synthesised 2-AG to promote axonal growth and guidance by autocrine activation of CB₁ receptors[18]. Neurons from AP-4-deficient mice exhibit axon-specific defects including reduced length and reduced branching[4,5]. Similarly, iPSC-derived neurons from patients with AP-4 deficiency also have reduced neurite length and reduced branching at an early stage of development[21]. Following our finding of low 2-AG levels in the brains of AP-4-deficient mice, we reasoned that this neurite outgrowth defect may arise due to DAGLB missorting and consequently impaired

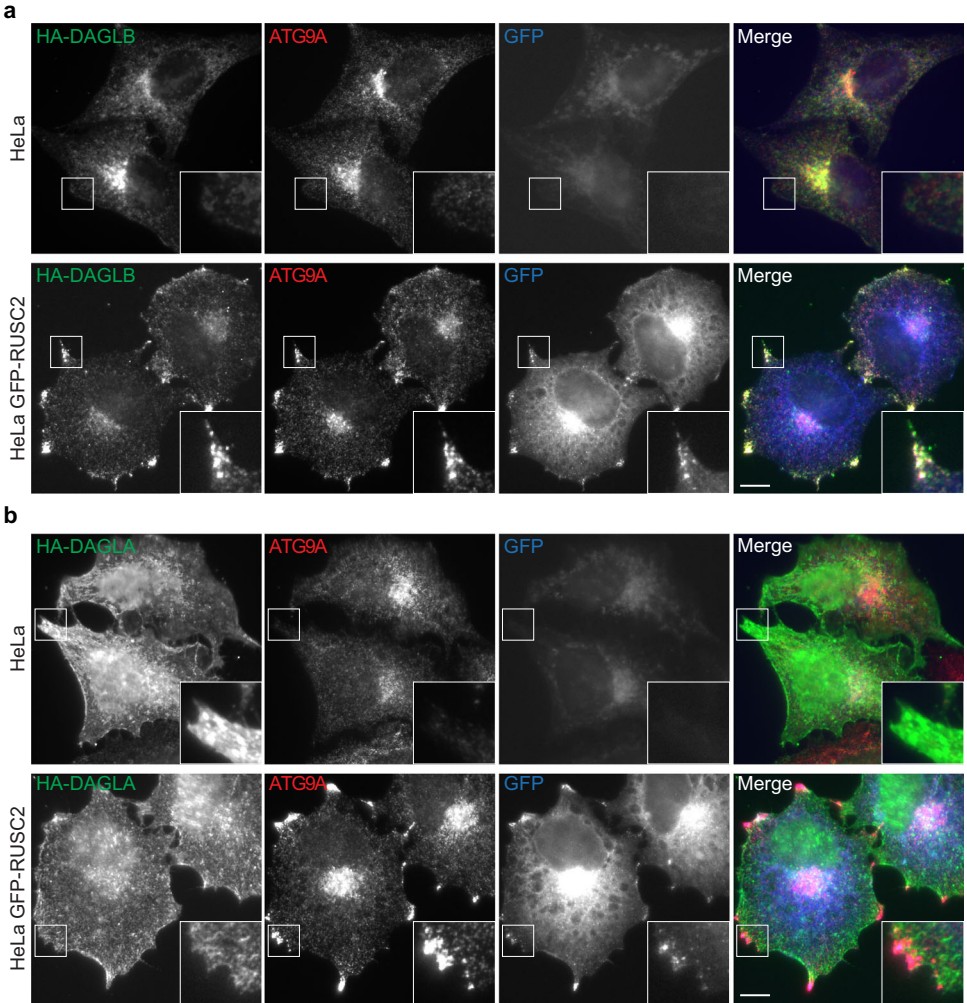

**Fig. 4 Unlike DAGLB, DAGLA is not rerouted in an AP-4 cargo redistribution assay. a** Widefield imaging of wild-type HeLa or HeLa cells stably expressing GFP-RUSC2 (blue), transiently expressing HA-DAGLB, and labelled with anti-HA (green) and anti-ATG9A (red). HA-DAGLB has a very similar localisation to ATG9A in both cell lines. The insets show the rerouting of HA-DAGLB to peripheral ATG9A- and RUSC2-positive puncta in cells overexpressing GFP-RUSC2. Scale bar: 10 μm. **b** Widefield imaging of wild-type HeLa or HeLa cells stably expressing GFP-RUSC2, transiently expressing HA-DAGLA, as in (**a**). HA-DAGLA has a different localisation to ATG9A and does not reroute to the peripheral puncta in cells overexpressing GFP-RUSC2. Scale bar: 10 μm. For (**a**) and (**b**), images are representative of at least 18 images per condition, including three independent transfections with microscopy performed independently for each.

synthesis of 2-AG. In this case, increasing the level of 2-AG in developing AP-4-deficient neurons may rescue the neurite out-growth phenotype. One way to increase the amount of 2-AG is to reduce its degradation by inhibiting MGLL, the enzyme responsible for 2-AG hydrolysis (Fig. 7a). *Mgll* knockout mice have tenfold elevated levels of brain 2-AG[31]. Likewise, pharmacological inhibition of MGLL results in significant elevations of 2-AG in cell lines and in vivo[32]. To test our hypothesis, we made use of a recently developed and highly selective MGLL inhibitor, ABX-1431[33]. First, we quantified the neurite outgrowth defect in iPSC-derived neurons from two different SPG47 patients (Fig. 7e, f and Supplementary Fig. 6b), using our previously developed live cell imaging assay[21]. For both patient lines, neurite outgrowth was significantly reduced compared to matched control neurons during the first 25 h post-plating. Next, we tested the effect of MGLL inhibition at different concentrations of ABX-1431. We found that low nanomolar concentrations of ABX-1431 (10 nM and 50 nM) increased the length of patient neurites to the length of healthy control neurites (Fig. 7g and Supplementary Fig. 6c). Importantly, at these concentrations neurite outgrowth of control neurons was not significantly affected (Fig. 7h and Supplementary

Fig. 6d). Hence, MGLL inhibition did not cause general neurite growth, but specifically rescued the AP-4 phenotype. We observed a similar rescue of the number of neurite branches in AP-4 patient neurons treated with ABX-1431 (Supplementary Fig. 7). Together, these data suggest that 2-AG levels are limiting for neurite outgrowth in AP-4-deficient iPSC neurons and indicate that MGLL inhibition could be a therapeutic avenue for AP-4 deficiency syndrome.

**A new cellular disease model for AP-4 deficiency syndrome.**
Despite the importance of DAGLs for axon development, the mechanism of their axonal targeting is currently unknown. AP-4 mediates axonal delivery of ATG9A[5]. Here we show that DAGLB is present in the same AP-4-derived vesicles as ATG9A (Figs. 3, 4 and 5), and accumulates at the TGN of iPSC-derived neurons from patients with AP-4 deficiency (Fig. 6a–c), while it is depleted from the axon (Fig. 6e, f). Therefore, our data strongly support that AP-4 is responsible for the axonal targeting of DAGLB (Fig. 8). Brain imaging studies of AP-4-deficient patients show features that suggest underdevelopment or loss of axons, in particular of the long projection neurons, including thinning of

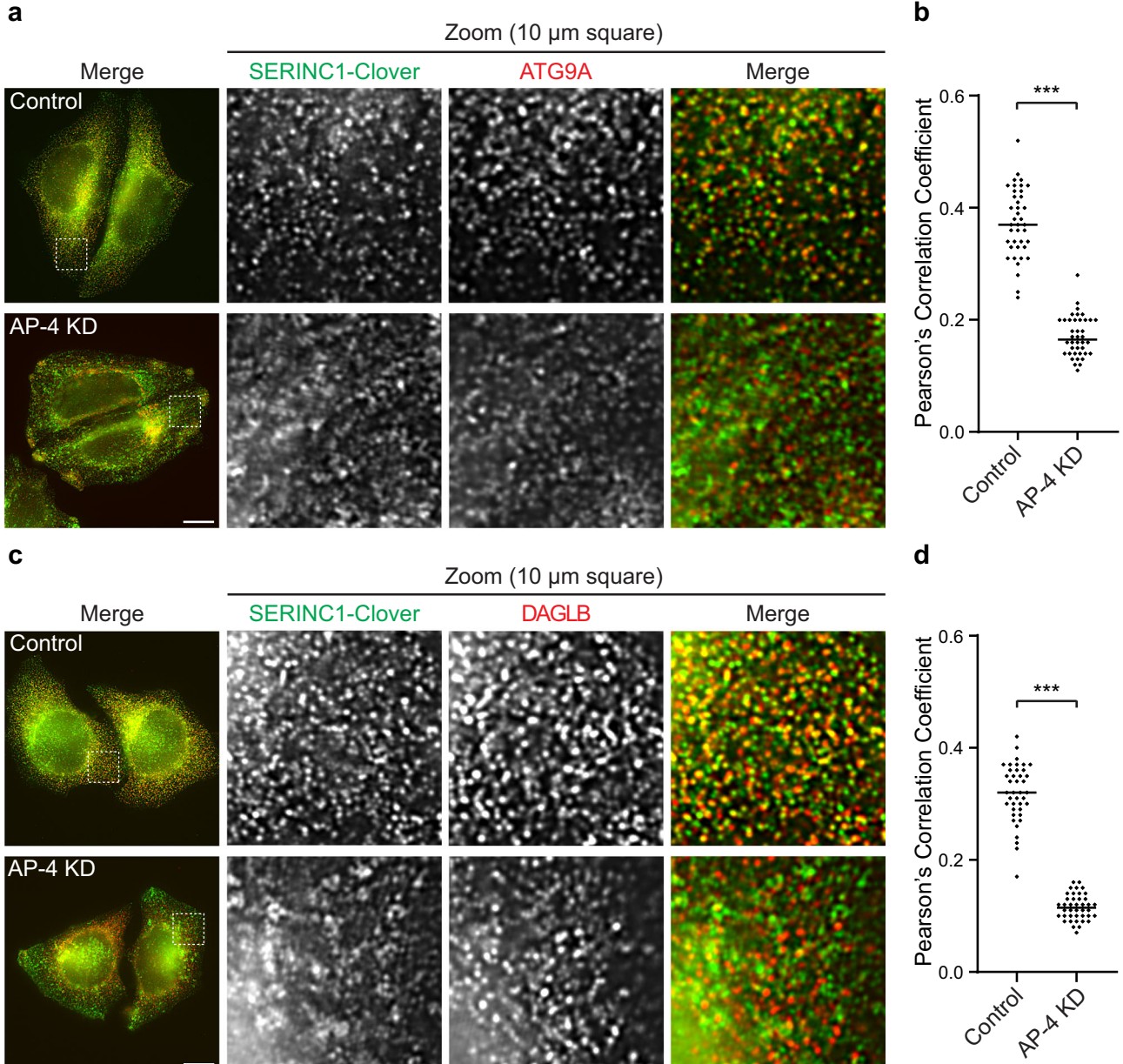

**Fig. 5 DAGLB and ATG9A colocalise with SERINC1 in an AP-4-dependent manner.** HeLa cells tagged endogenously with Clover (modified GFP) at the C-terminus of SERINC1 were transfected with siRNA to knock down (KD) AP-4, or were transfected with a non-targeting siRNA (Control). **a** Super-resolution structured illumination microscopy (SR-SIM) was used to image SERINC1-Clover (via anti-GFP; green) and anti-ATG9A (red). Representative images show the whole field of view and a zoomed image of a peripheral $10 \times 10 \ \mu m^2$ square. ATG9A and SERINC1 colocalised in small puncta throughout the cytoplasm in control cells, but not in AP-4 depleted cells. Scale bar: 10 μm. **b** Quantification of colocalisation between SERINC1-Clover and ATG9A in control and AP-4 knockdown (KD) cells, using Pearson's Correlation Coefficient (PCC). The experiment was performed in biological duplicate and the graph shows combined replicate data: each datapoint indicates the PCC for an individual cell (horizontal bar indicates median; $n = 40$ cells per condition, examined across 2 independent experiments). Data were subjected to a two-tailed Mann–Whitney $U$-test: ***$p \leq 0.001$ ($p = 9.3 \times 10^{-23}$). **c** SR-SIM was used to image SERINC1-Clover (via anti-GFP; green) and anti-DAGLB (red), as in (**a**). Like ATG9A, DAGLB colocalised with SERINC1 in small puncta throughout the cytoplasm in control cells, but not in AP-4 depleted cells. Scale bar: 10 μm. **d** Quantification of colocalisation between SERINC1-Clover and DAGLB in control and AP-4 KD cells, using PCC. The experiment was performed in biological duplicate and the graph shows combined replicate data: each datapoint indicates the PCC for an individual cell (horizontal bar indicates median; $n = 40$ cells per condition, examined across 2 independent experiments). Data were subjected to a two-tailed Mann–Whitney $U$-test: ***$p \leq 0.001$ ($p = 1.9 \times 10^{-23}$). Source data are provided as a Source Data file.

the corpus callosum and ventriculomegaly (enlarged ventricles)[11,34]. On the other hand, endocannabinoid signalling is required for elongation and fasciculation of the long axons of pyramidal cells (a common class of projection neurons in the cerebral cortex)[30]. Both pharmacological inhibition and genetic ablation of $CB_1R$ cause defects in axonal pathfinding[30,35,36]. Thus, we propose that dysregulation of the spatial control of 2-AG

production may contribute to developmental axon defects in AP-4 deficiency syndrome (Fig. 8). This model is fully supported by our MGLL inhibition experiment, as impaired neurite outgrowth of AP-4-deficient neurons is specifically rescued to wild-type levels by the MGLL inhibitor ABX-1431 (Fig. 7e–h).

MGLL links the endocannabinoid and prostaglandin signalling networks via the generation of AA, a precursor of inflammatory

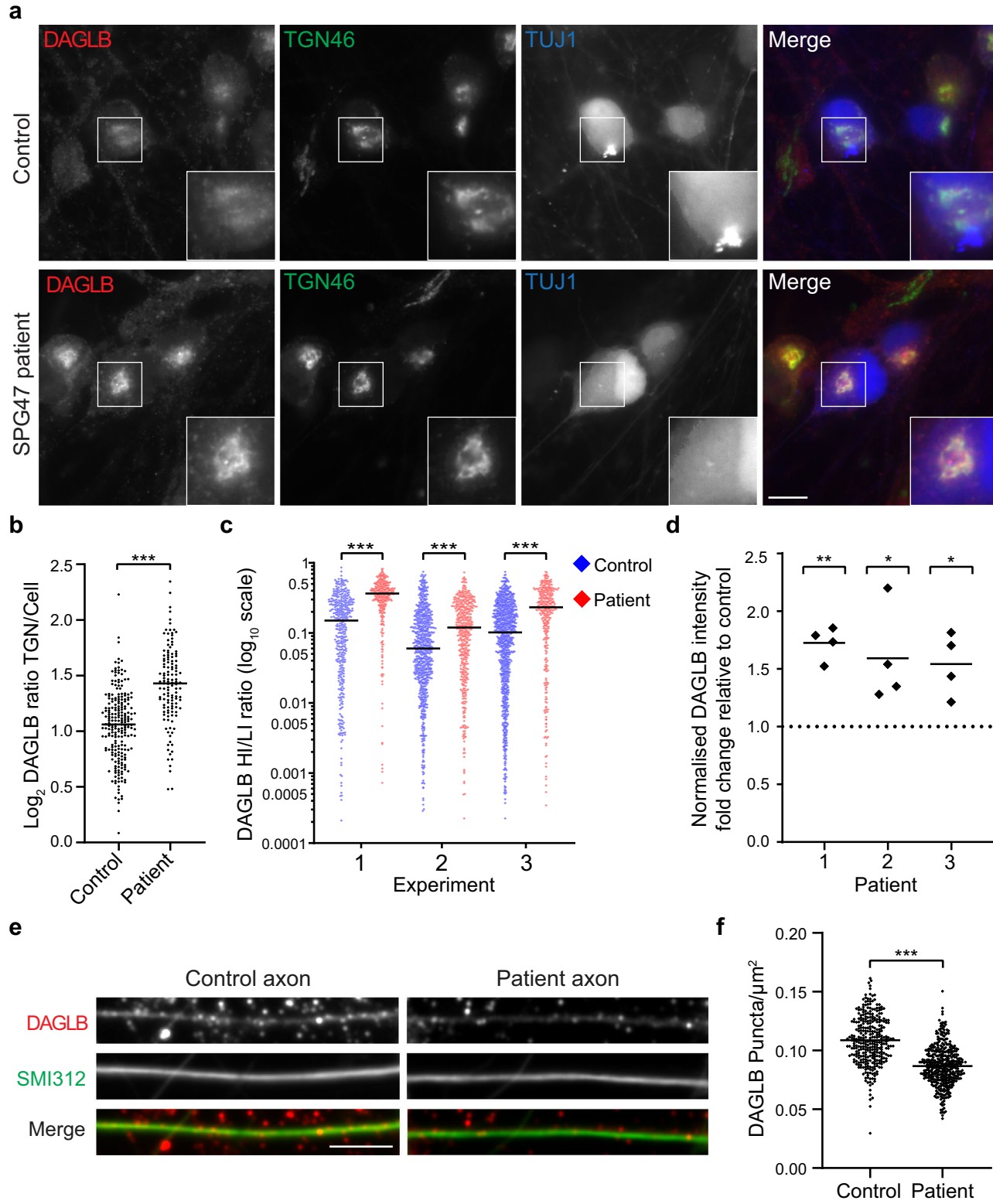

prostaglandins[37]. Given its central position in the regulation of these important lipid signalling pathways, MGLL is an attractive target for the treatment of diverse pathologies including neurodegeneration, neuropathic pain, cancer, inflammation, and metabolic disorders[32,38]. ABX-1431 is a highly specific MGLL inhibitor that was identified by activity-based protein profiling[33]. It is orally available and CNS penetrant, with a reported IC50 of 14 nM, and in clinical trials for several neurological disorders,

including phase I trials for neuropathic pain and inflammatory CNS disorders as well as a phase II trial for Tourette syndrome[32]. It will now be important to assess the efficacy of ABX-1431 in further pre-clinical models of AP-4 deficiency syndrome, for example, AP-4-deficient zebrafish, which also exhibit an axon growth defect[39]. Our data suggest that the level of MGLL inhibition must be finely tuned, as complete rescue of neurite length only occurred at lower concentrations of ABX-1431

**Fig. 6 DAGLB is missorted in iPSC-derived cortical neurons from AP-4-deficient patients.** iPSCs from patients with *AP4B1*-associated AP-4 deficiency syndrome (SPG47) and their unaffected same sex heterozygous parents (control) were differentiated into cortical neurons. **a** Widefield imaging of immunofluorescence triple labelling of DAGLB (red), TGN46 (green) and TUJ1 (a marker to distinguish neurons from co-cultured astrocytes; blue) in iPSC neurons from patient 1 and their matched control. DAGLB signal was increased at the trans-Golgi network (TGN) in the AP-4 patient cells. Scale bar: 10 μm. **b** Quantification of the ratio of DAGLB labelling intensity between the TGN and the rest of the cell, in the cells shown in (**a**). The experiment was performed in technical triplicate and the graph shows combined replicate data: each datapoint indicates the $\log_2$ ratio for an individual cell (horizontal bar indicates median; $n = 227$ cells for control; $n = 129$ cells for patient). Data were subjected to a two-tailed Mann–Whitney *U*-test: ***$p \leq 0.001$ ($p = 6.7 \times 10^{-20}$). **c** High-throughput confocal imaging was used to assay the distribution of DAGLB in iPSC-derived neurons from patient 1 and their matched control. Neurons in 96-well plates were labelled with antibodies against DAGLB, GOLGA1 (a TGN marker) and TUJ1. The ratio between the area of high intensity (HI; overlaps with TGN) and low intensity (LI) DAGLB labelling was quantified from three differentiations per cell line (biological triplicate; plotted separately): each datapoint indicates the ratio for an individual cell, plotted on a $\log_{10}$ scale (horizontal bar indicates median; experiment 1/2/3: $n = 417/786/999$ cells for control; $n = 306/464/339$ cells for patient). Data were subjected to a two-tailed Mann–Whitney *U*-test for comparison of the patient and control within each differentiation: ***$p \leq 0.001$ (1: $p = 2.4 \times 10^{-37}$; 2: $p = 7.2 \times 10^{-10}$; 3: $p = 3.2 \times 10^{-23}$). Comparable results for another SPG47 patient (patient 2) are shown in Supplementary Fig. 5b. **d** Western blotting was used to quantify the level of DAGLB in whole-cell lysates from iPSC-derived neurons from three patients with SPG47 and their unaffected same sex heterozygous parents. Data are from $n = 4$ differentiations per cell line, and the graph shows fold change in normalised DAGLB intensity in the patient relative to control (horizontal bars indicate mean). Log-transformed absolute normalised intensity values were subjected to a two-tailed paired *t* test for comparison of each patient with their matched control: **$p \leq 0.01$; *$p \leq 0.05$ (1: $p = 0.001$; 2; $p = 0.036$; 3: $p = 0.019$). **e** High-throughput confocal imaging was used to assay the density of DAGLB puncta in axons of iPSC neurons from patient 1 and their matched control. Representative images are shown of anti-DAGLB (red) in axons marked with the axonal marker antibody cocktail SMI312 (green). Scale bar: 10 μm. **f** Quantification of the number of DAGLB puncta per area (μm$^2$) in axons from patient 1 and their matched control, from high-throughput confocal images as shown in (**e**). Each datapoint represents the mean number of DAGLB puncta per axon area per image ($n = 324$ images in the control group and $n = 405$ images in the patient group, covering 21902 and 25253 axon segments respectively). Data were subjected to a two-tailed unpaired *t* test: ***$p \leq 0.001$ ($p = 3.7 \times 10^{-50}$). Source data are provided as a Source Data file.

(Fig. 7g and Supplementary Fig. 6c). The reason for this is currently unclear, but chronic MGLL inhibition has been shown to result in functional antagonism of the endocannabinoid system[31].

Our study has revealed an intriguing difference between the trafficking of DAGLB and DAGLA (Fig. 4). While DAGLB is trafficked by AP-4 vesicles in diverse cell types, DAGLA does not appear to be an AP-4 cargo. Although DAGLB is closely related to DAGLA, DAGLA has an additional C-terminal tail (~370 amino acids) that is not present in DAGLB[17]. The DAGLA tail contains a motif for binding the Homer family of adaptor proteins, which is suggested to have a role in either targeting or retaining the enzyme at the postsynaptic density[40]. This aligns with the important role for DAGLA in retrograde synaptic signalling, for which DAGLB is dispensable[28,41]. Our data now provide further evidence for differential sorting of DAGLA and DAGLB, and it will be important to investigate the sequence-encoded determinants of this specificity. In addition, DAGL subcellular localisation is highly regulated during development, switching from axonal in the embryonic brain to predominantly dendritic in the adult brain[17,36]. Due to the lack of a knockout-validated antibody for murine DAGLB[40], previous developmental studies have largely focused on DAGLA. Our discovery of a specific role for AP-4 in axonal targeting of DAGLB encourages further investigation into the spatial and developmental regulation of the two DAGLs. Of note, there is a rich field of research that has characterised the intricate spatial coordination of components of the endocannabinoid signalling network, including DAGLs and MGLL, as well as CB₁R, which has been observed in small axonal vesicles[42]. The coincident axonal targeting of DAGLs and CB₁R in developing neurons raises the question whether AP-4 might also have a role in axonal transport of CB₁R, which should be addressed in future studies.

In conclusion, we have identified an important role for AP-4 in transport of DAGLB, and thus provide a missing link in understanding the spatial regulation of endocannabinoid signalling. Our findings open up a new direction for the pathomechanisms of AP-4 deficiency syndrome: aberrant spatial control of 2-AG synthesis leading to developmental defects in axon formation. We suggest that the diverse neurological symptoms of AP-4-deficient patients may be caused by compound effects of impaired 2-AG production due to DAGLB missorting and impaired autophagy due to ATG9A missorting, impacting on axonal development and axonal maintenance, respectively (Fig. 8). Moreover, the occurrence of an autophagy-independent axon growth defect in *Atg9a* knockout neurons[14] suggests that there may be further connections between DAGLB and ATG9A. While DAGLB has not been linked to autophagy, ATG9A is a lipid scramblase[6,7] and has functions independent of autophagy[14,43,44]. This raises the intriguing possibility that ATG9A could also have a role in controlling access of DAGLB to its substrate, or in distributing its product 2-AG. Finally, it will be important to test if increasing 2-AG levels through MGLL inhibition might help to correct the axonal defects observed in pre-clinical models of AP-4 deficiency, which may enable new treatment strategies for AP-4 deficiency syndrome.

## Methods

**Ethics statement**. All animal procedures were conducted at the University of Cambridge, in accordance with the UK Animals (Scientific Procedures) Act of 1986 and under the authority of the UK Home Office Project Licence PPL 70/8339 held by Prof. Margaret Robinson, and were approved by the UK Home Office and the University of Cambridge Animal Welfare and Ethical Review Committee. The recruitment and generation of the human iPSC lines used in this study were previously described[45]. The protocol for generating human iPSCs was approved by the Institutional Review Board at Boston Children's Hospital (IRB#: P00016119). Written consent was obtained from all probands and/or their legal guardian. No compensation was provided.

**Animals**. *Ap4e1* knockout mice (C57Bl/6N-Ap4e1[tm1b{KOMP}Wtsi]), generated using the knockout-first tm1b allele system[46], were obtained via the European Mouse Mutant Archive (EMMA; strain EM:09451) from Helmholtz Zentrum München Deutsches Forschungszentrum für Gesundheit und Umwelt (GmbH). Mice were housed at an ambient temperature of 19–23 °C, recorded daily, with a humidity of 55% (±10%). The light/dark cycle was 7 am (lights on) to 7 pm (lights off), with dawn and dusk periods, and animals had free access to food (SAFE® A05 complete long-term diet from SAFE® Complete Care Competence) and water. For genotyping, genomic DNA was isolated from ear snips using a High Pure PCR Template Purification Kit (Roche). PCR was performed using the following primers: AP4E1-5arm-WTF (5′-GCCTCTGTTTAGTTTGCGATG-3′); AP4E1-Crit-WTR (5′-CGTGCACAGACAGGTTTGAT-3′); 5mut-R1 (5′-GAACTTCGGGAAT AGGAACTTCG-3′). This yielded a 271 bp fragment from the wild-type allele and a 130 bp fragment from the knockout allele. The brain samples used in this study

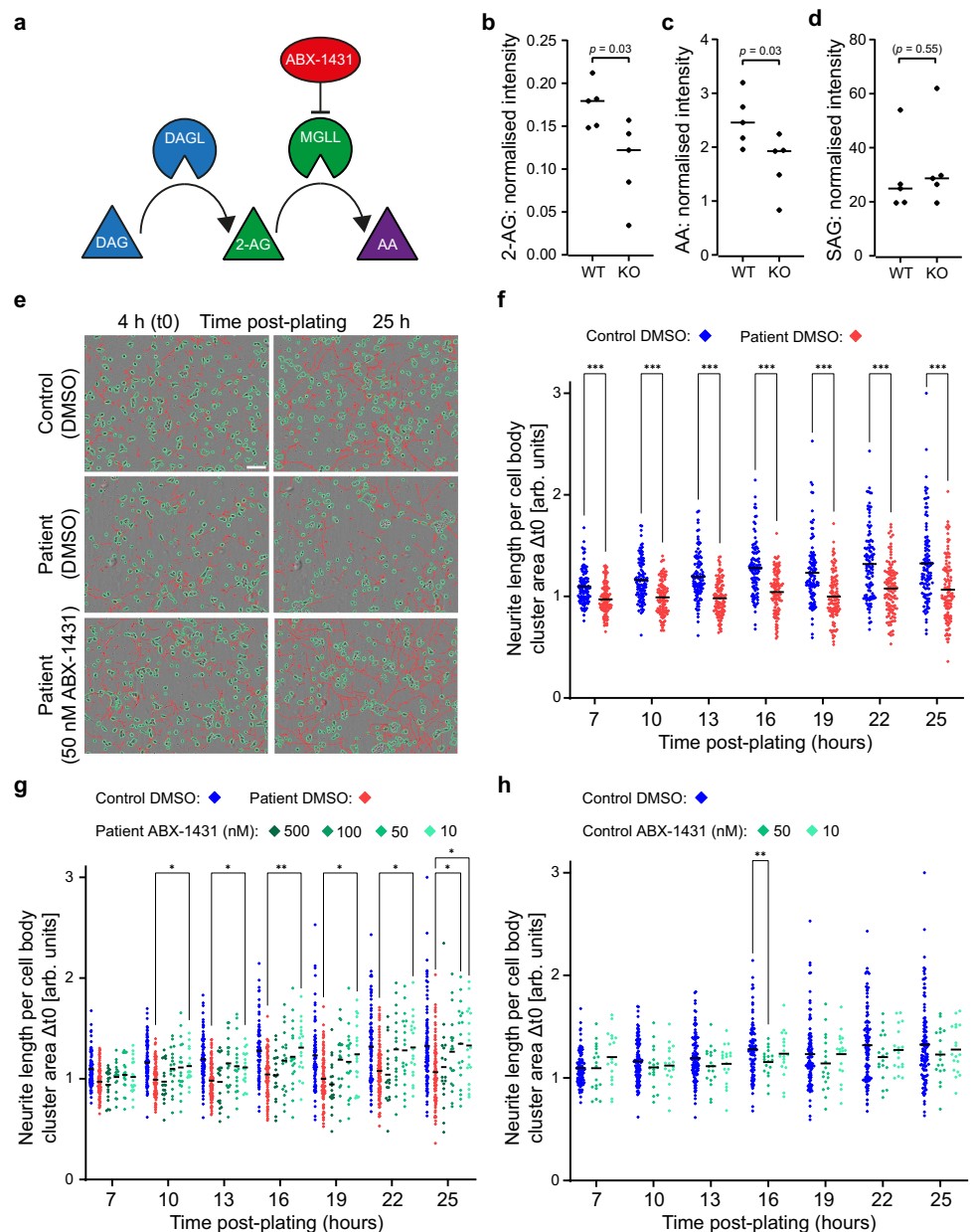

were taken at 6 months (range 178–197 days). The WT group consisted of 4 females and 1 male; the KO group consisted of 3 females and 2 males.

**Antibodies and reagents**. The following antibodies were used in this study: mouse anti-alpha tubulin 1:1000 for WB (clone DM1A, Sigma-Aldrich Cat# T9026, RRID:AB_477593), rabbit anti-alpha tubulin 1:1000 for WB (Abcam Cat# ab18251, RRID:AB_2210057), rabbit anti-AP4E1[47] 1:1000 for WB (a gift from Margaret Robinson, University of Cambridge), rabbit anti-ATG9A 1:1000 for WB and 1:100 for IF (clone EPR2450(2), Abcam Cat# ab108338, RRID:AB_10863880), guinea pig anti-beta tubulin III (TUJ1) 1:500 for IF (Synaptic Systems Cat# 302 304, RRID:AB_10805138), mouse anti-beta tubulin III (TUJ1) 1:800 for IF (clone SDL.3D10, Sigma-Aldrich Cat# T8660, RRID:AB_477590), rabbit anti-DAGLB 1:1000 for WB and 1:800 for IF (Abcam Cat# ab191159), chicken anti-GFP 1:500 for IF (Abcam Cat# ab13970, RRID:AB_300798), mouse anti-Golgin 97 (GOLGA1) 1:500 for IF (Abcam Cat# ab169287), mouse anti-HA 1:500 for IF (clone 16B12, BioLegend Cat# 901501, RRID:AB_2565006), mouse anti-Neurofilament (SMI312) 1:1000 for IF (BioLegend Cat# 837904, RRID:AB_2566782) and sheep anti-TGN46 1:200 for IF (Bio-Rad Cat# AHP500, RRID:AB_324049). Secondary antibodies used for WB were: Horseradish peroxidase (HRP)-conjugated goat anti-mouse IgG (Cat# AP308P, RRID:AB_11215796) and goat anti-rabbit IgG (Cat# AP307P, RRID:AB_11212848), purchased from Sigma-Aldrich and used at 1:5000, and near-infrared fluorescent-labelled secondary antibodies IRDye 680LT donkey anti-mouse IgG (IR680LT, Cat# 926-68022,

RRID:AB_10715072) and IRDye 800CW donkey anti-rabbit IgG (IR800CW, Cat# 926-32213, RRID:AB_621848), purchased from LI-COR Biosciences and used at 1:10000. Fluorescently labelled secondary antibodies used in this study were Alexa Fluor Plus 488-labelled goat anti-chicken IgY (Cat# A32931, RRID:AB_2762843), Alexa Fluor 488-labelled goat anti-guinea pig IgG (Cat# A-11073, RRID:AB_2534117), Alexa Fluor 488-labelled goat anti-mouse IgG (Cat# A-11029, RRID:AB_2534088), Alexa Fluor 594-labelled goat anti-mouse IgG (Cat# A11005, RRID:AB_2534073), Alexa-Fluor Plus 680-labelled donkey anti-mouse IgG (Cat# A32788, RRID:AB_2762831), Alexa Fluor 568-labelled donkey anti-rabbit IgG (Cat# A10042, RRID:AB_2534017), Alexa Fluor 647-labelled goat anti-rabbit IgG (Cat# A21245, RRID:AB_2535813), and Alexa Fluor 680-labelled donkey anti-sheep IgG (Cat# A-21102, RRID:AB_2535755), all purchased from Thermo Fisher Scientific and used at 1:500. The specificity of antibodies against the key protein targets of this study (human DAGLB & ATG9A) was confirmed in-house using siRNA-mediated knockdown in HeLa cells, by Western blotting (loss of band of expected size) and immunofluorescence microscopy (loss of fluorescent signal). The anti-ATG9A antibody is KO-validated by the manufacturer (Abcam). Fluorescently labelled secondary antibodies were confirmed to be specific through no primary antibody controls. The other primary antibodies, used as cellular markers, were not validated.

ABX-1431 (MedChemExpress Cat# HY-117632), a hexafluoro isopropyl carbamate-derived, covalent, irreversible monoacylglycerol lipase inhibitor[33], was prepared in DMSO (American Bioanalytical Cat# AB03091-00100).

**Fig. 7 2-AG and AA are reduced in AP-4 knockout brains and MGLL inhibition rescues impaired neurite outgrowth in AP-4-deficient neurons. a** Diagram of the 2-AG biosynthesis pathway. Hydrolysis of diacylglycerol (DAG; blue) by DAG lipase (DAGL; blue) generates 2-arachidonoylglycerol (2-AG; green), which is hydrolysed by monoacylglycerol lipase (MGLL; green) to generate arachidonic acid (AA; purple). MGLL inhibition by the specific inhibitor ABX-1431 (red) blocks 2-AG hydrolysis and thereby increases the level of 2-AG[33]. **b–d** Mass spectrometry-based quantification of (**b**) 2-AG, (**c**) AA and (**d**) 1-stearoyl-2-arachidonoyl-sn-glycerol (SAG), from wild-type (WT) and *Ap4e1* knockout (KO) mouse brains ($n = 5$ animals per group; horizontal bar indicates median; mice were aged 6 months). Data were subjected to a two-tailed Mann–Whitney $U$-test. **e** Neurite outgrowth was assayed in iPSC-derived cortical neurons from a patient with *AP4B1*-associated AP-4 deficiency syndrome (SPG47; patient 1) and their unaffected same sex heterozygous parent (control), using automated live cell imaging. Neurons were cultured in the presence of DMSO (vehicle control) or the MGLL inhibitor ABX-1431 at 10, 50, 100 or 500 nM (the highest two doses were administered only to the patient neurons). Neurons were monitored from 4 h post-plating, with images captured every 3 h until 25 h post-plating. Representative images are shown at 4 h (t0) and 25 h post-plating, with cell bodies outlined in green and neurites traced in red. Scale bar: 100 μm. **f–h** Automated image analysis from neurite outgrowth assay of two separate neuronal differentiations per cell line shown in (**e**). Graphs show neurite length per image over time, normalised to cell body cluster area. Data are shown relative to normalised neurite length at t0. The same data are shown for the control plus DMSO condition in (**f–h**) and for the patient plus DMSO condition in (**f**) and (**g**). **f** Average neurite length of patient neurons was significantly reduced compared to control neurons at all time points. Per group, $n = 108$ images from two biological replicates were analysed. Data were subjected to a two-way repeated measures ANOVA with Šídák's multiple comparisons test for comparisons at each time point between control and patient: ***$p \leq 0.001$. **g** Average neurite length of patient neurons was rescued by treatment with 10 nM ABX-1431. 50 and 100 nM ABX-1431 doses also increased neurite length, whereas neurite length was not improved by the highest dose (500 nM). Per treatment group, $n = 18$ images from two biological replicates were analysed. Data were subjected to a two-way repeated measures ANOVA with Dunnett's multiple comparisons test for comparisons at each time point between each dose of ABX-1431 and the patient plus DMSO control: **$p \leq 0.01$; *$p \leq 0.05$. The control plus DMSO condition is shown for reference, but was not included in the statistical analysis. **h** Average neurite length of control neurons was not affected by treatment with 10 or 50 nM ABX-1431. Per treatment group, $n = 18$ images from two biological replicates were analysed. Data were subjected to a two-way repeated measures ANOVA with Dunnett's multiple comparisons test for comparisons at each time point between each dose of ABX-1431 and the DMSO control: *$p \leq 0.05$. In (**f–h**) only significant changes are annotated; all other comparisons resulted in non-significant $p$ values ($p > 0.05$). Source data are provided as a Source Data file.

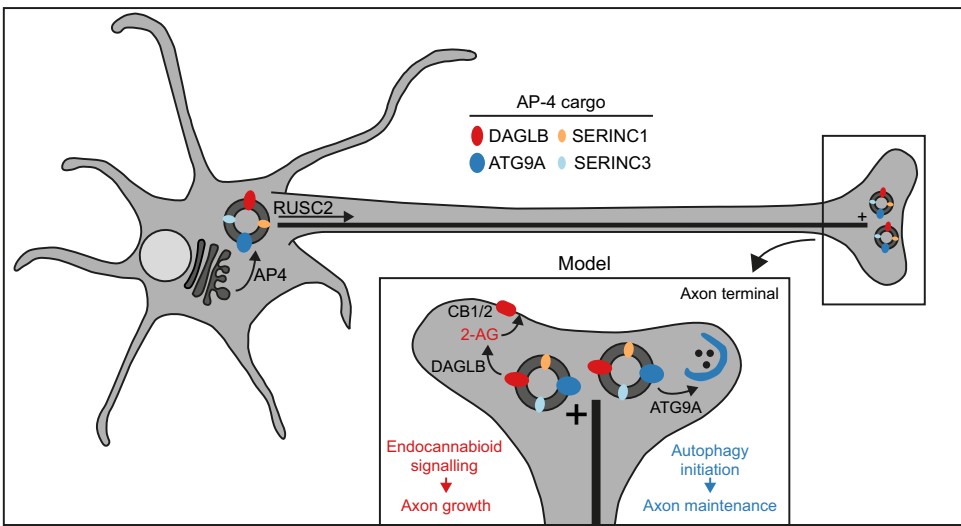

**Fig. 8 A new cellular disease model for AP-4 deficiency syndrome.** Proposed model for the role of AP-4 in axonal transport based on our new and published data. AP-4 acts at the trans-Golgi network (TGN) membrane to package its cargo proteins, DAGLB, ATG9A, SERINC1 and SERINC3, into transport vesicles. RUSC2 mediates microtubule plus-end-directed transport of AP-4-derived vesicles, delivering cargo to the distal axon. DAGLB is an enzyme responsible for production of the endocannabinoid 2-AG, known to be required for axonal growth via autocrine activation of cannabinoid receptors. The previous model for AP-4 deficiency syndrome has focused on missorting of ATG9A, which is required for autophagy initiation and hence for axonal maintenance. We now propose that the neuronal pathology in AP-4 deficiency arises from the compound effects of DAGLB missorting on axonal development and ATG9A missorting on axonal autophagy.

**Cell culture**. HeLa M cells[48] were a gift from Paul Lehner, University of Cambridge. Other cell lines used in this study were made in our lab and reported in previous publications: *AP4B1* knockout HeLa (clone x2A3)[49]; wild-type AP4B1-rescued *AP4B1* knockout HeLa[49]; *AP4E1* knockout HeLa (clone x6C3)[2]; CRISPR-Cas9 AP4B1-depleted SH-SY5Y, CRISPR-Cas9 AP4E1-depleted SH-SY5Y, and their Cas9-expressing parental cells (mixed populations)[2]; wild-type HeLa and *AP4B1* knockout HeLa stably expressing RUSC2-GFP[2]; *AP4E1* knockout HeLa stably expressing GFP-RUSC2[2]; endogenously-tagged HeLa SERINC1-Clover (clone A3) and HeLa SERINC3-Clover (clone B6)[2]. For HA-DAGLA and HA-DAGLB localisation experiments, a previously described HeLa cell line stably expressing GFP-tagged RUSC2 (clone 3)[2] was re-single cell cloned for uniform expression to give HeLa GFP-RUSC2 clone D4.

HeLa and SH-SY5Y cells were maintained in Dulbecco's Modified Eagle's Medium (DMEM) high glucose supplemented with 10% v/v foetal calf serum, 4 mM

l-glutamine, 100 U mL$^{-1}$ penicillin and 100 μg mL$^{-1}$ streptomycin and all cells were cultured at 37 °C under 5% CO$_2$. Differentiation of SH-SY5Y cells into a neuron-like state was by sequential culture in the presence of retinoic acid and BDNF[50]. Cells were seeded at 25% confluency in standard culture medium and the following day the medium was replaced with medium containing 10 μM all-trans-Retinoic acid (Sigma-Aldrich Cat# R2625; 10 mM stock in DMSO). Two days later cells were split 1:4 and cultured for 3 further days in the presence of retinoic acid (5 days total). Cells were then washed in PBS and cultured for 5 days in serum-free DMEM containing 50 ng mL$^{-1}$ BDNF (Miltenyi Biotec Cat# 130-096-285; 50 μg mL$^{-1}$ stock in sterile ddH$_2$O).

No cell lines used in this study were found in the database of commonly misidentified cell lines that is maintained by ICLAC and NCBI Biosample. The cell lines were routinely tested for the presence of mycoplasma contamination using DAPI to stain DNA and a PCR-based mycoplasma test (PanReac AppliChem ITW Reagents Cat# A3744).

**iPSC-derived cortical neurons**. iPSCs from individuals with AP4B1-associated SPG47 (AP4B1, NM_001253852.3 [https://www.ncbi.nlm.nih.gov/nuccore/NM_001253852.3/]): Patient 1 (BCHNEUi001-A; male, age 2 years), c.1345A>T (p.Arg449Ter)/c.1160_1161del (p.Thr387ArgfsTer30); patient 2 (BCHNEUi005-A; female, age 3 years 9 months), c.1216C>T (p.Arg406Ter)/c.1328T>C (p.Leu443-Pro); patient 3 (BCHNEUi003-A; female, age 3 years), c.530_531insA (p.Asn178-GlufsTer20)/c.114-2A>C) and their sex-matched parents (heterozygous carriers, clinically unaffected) were generated previously in our lab[45], and were maintained and differentiated into neurons as described here and in published studies[21,45]. iPSCs were maintained in StemFlex medium (Thermo Fisher Scientific Cat# A3349401) on Geltrex-coated plates (Thermo Fisher Scientific Cat# A1413202) and passaged weekly with Gentle Cell Dissociation Reagent (STEMCELL Technologies Cat# 07174). Cortical neurons were differentiated using induced NGN2 expression according to protocols modified from Zhang et al.[51] Briefly, human iPSCs were dissociated into single cells with Accutase (Innovative Cell Technology Cat# AT104-500) and seeded onto Geltrex-coated plates. On differentiation day -1, iPSCs were infected with concentrated rtTA-, EGFP-, and NGN2-expressing lentiviruses in presence of polybrene (Sigma-Aldrich Cat# TR-1003-G). The next day, NGN2 expression was induced using doxycycline (Millipore Cat# 324385), and infected cells were then selected using puromycin (Thermo Fisher Scientific Cat# A1113803) for up to 48 h. Growth factors BDNF (10 ng mL$^{-1}$, Peprotech Cat# 450-02) and NT3 (10 ng mL$^{-1}$, Peprotech Cat# 450-03), and laminin (0.2 μg L$^{-1}$, Thermo Fisher Scientific Cat# 23017-015) were added in DMEM/F-12 medium with GlutaMAX, supplemented with N2 and nonessential amino acids for the first 2 days. Cells were then fed every other day with Neurobasal-A medium supplemented with B27 and GlutaMAX, containing BDNF (10 ng mL$^{-1}$), NT3 (10 ng mL$^{-1}$), laminin (0.2 μg L$^{-1}$), doxycycline (2 μg mL$^{-1}$), and Ara-C (2 μM, Sigma-Aldrich Cat# C1768), until dissociation on day 6. On day 6, cells were dissociated with papain (Worthington Cat# LK003178) supplemented with DNase (Worthington Cat# LK003172). For experiments shown in Fig. 6a–c, e, f and Supplementary Fig. 5a, b, neurons were replated with human iPSC-derived astrocytes (Astro.4U; Ncardia) onto poly-D-lysine and laminin-coated plates. Cultures were then fed every other day with Neurobasal-A medium supplemented with B27, GlutaMAX, transferrin, 45% glucose, 8% NaHCO$_3$, BDNF (10 ng mL$^{-1}$), NT3 (10 ng mL$^{-1}$) and laminin (0.2 μg L$^{-1}$). For experiments shown in Fig. 6d and Supplementary Fig. 5c, neurons were replated onto poly-D-lysine and laminin-coated plates without astrocytes and fed every other day with astrocyte-derived conditioned media, Neurobasal-A medium supplemented with B27, GlutaMAX, 45% glucose, 8% NaHCO$_3$, transferrin, BDNF (10 ng mL$^{-1}$), NT3 (10 ng mL$^{-1}$) and laminin (0.2 μg L$^{-1}$). For assessment of neurite outgrowth and the effects of MGLL inhibitor treatment shown in Fig. 7e–h, Supplementary Fig. 6b–d and Supplementary Fig. 7, neurons were replated onto poly-D-lysine and laminin coated plates without astrocytes and treated with compounds as described below.

**Transfections and siRNA-mediated knockdown**. Transient DNA transfections were carried out using a TransIT-HeLaMONSTER® kit (Mirus Bio LLC), according to the manufacturer's instructions. Wild-type AP4B1 was expressed using a previously described construct, pLXIN_AP4B1[49]. HA-hDAGLa-V5 and HA-hDAGLb-V5 were a gift from Pat Doherty (Addgene plasmid #87674[26] [http://n2t.net/addgene:87674]; RRID:Addgene_87674).

Knockdown of AP-4 was achieved by combined siRNA targeting of AP4E1 and AP4M1 using ON-TARGETplus SMARTpools (AP4E1, L-021474-00; AP4M1, L-011918-01; Dharmacon), using a double-hit 96 h protocol[15]. For the first hit the final concentration of siRNA was 40 nM (20 nM AP4M1 + 20 nM AP4E1). The second hit was performed 48 h after the first hit with half the final concentration of siRNA. Transfections of siRNA were carried out with Oligofectamine (Thermo Fisher Scientific), according to the manufacturer's instructions and control cells were transfected with ON-TARGETplus Non-targeting siRNA #1 (D-001810-01, Dharmacon).

**Western blotting**. Estimations of protein concentrations were made using a Pierce BCA Protein Assay Kit (Thermo Fisher Scientific Cat# 23225). Cells were lysed for Western blot analysis in 2.5% (w/v) SDS/50 mM Tris pH 8 and incubated at 65 °C for 3 min. Lysates were then sonicated in a Bioruptor (fifteen 30 seconds on/off cycles at maximum intensity) to ensure complete solubilisation and fragmentation of nucleic acids. Samples were heated at 72 °C for 10 min in NuPAGE LDS Sample Buffer (Thermo Fisher Scientific Cat# NP0007). Samples were loaded at equal protein amounts for SDS-PAGE, performed on NuPAGE 4–12% Bis–Tris gels in NuPAGE MOPS SDS Running Buffer (all Thermo Fisher Scientific). PageRuler Plus Prestained Protein Ladder (Thermo Fisher Scientific) was used to estimate the molecular size of bands. For Western blots shown in Fig. 2e, f, proteins were transferred to nitrocellulose membrane by wet transfer and membranes were blocked in 5% w/v milk in PBS (137 mM NaCl, 2.7 mM KCl, 10 mM Na$_2$HPO$_4$ and 1.76 mM KH$_2$PO$_4$, pH 7.4) with 0.1% v/v Tween-20 (PBS-T). Primary antibodies (diluted in 5% BSA in PBS-T) were added for at least 1 h at room temperature, followed by washing in PBS-T, incubation in secondary antibody (in 5% milk) for 30 min at room temperature, washing in PBS-T and finally PBS. Chemiluminescence detection of HRP-conjugated secondary antibody was carried out using Amersham ECL Prime Western Blotting Detection Reagent (GE Healthcare) and

the ImageQuant LAS 4000 imaging system (GE Healthcare). Where representative blots are shown, the experiment was repeated at least twice. Quantification was carried out using the Gel Analyzer function of ImageJ[52] version 2.1.0. Data were from four blots per cell line and three separate differentiations. DAGLB absolute intensity values were normalised using corresponding absolute intensity values for alpha-tubulin, as a loading control. The absolute normalised intensity values were log$_{10}$-transformed and analysed using a one-way ANOVA with Dunnett's Multiple Comparison Test for comparisons to the control.

For Western blots shown in Fig. 6d and Supplementary 5c, proteins were transferred to PVDF membrane (EMD Millipore Cat# IPFL00010) by wet transfer. Following blocking with blocking buffer (LI-COR Biosciences Cat# 927-60003) for 1 h at room temperature, membranes were incubated overnight with primary antibody. Near-infrared fluorescent-labelled secondary antibodies (IR800CW, IR680LT; LI-COR Biosciences Cat# 926-32213, 926-68022) were used for detection and quantification was carried out using the Odyssey CLx imaging system and Image Studio Lite Software (LI-COR Biosciences). Four independent experiments (separate differentiations) were performed for each cell line. DAGLB absolute intensity values were normalised using corresponding absolute intensity values for alpha-tubulin, as a loading control. The absolute normalised intensity values were log$_{10}$-transformed and analysed using a two-tailed paired t test for comparison of each patient with their matched control.

**Immunofluorescence labelling**. For widefield microscopy, cells were grown on 13 mm glass coverslips. For super-resolution structured illumination microscopy (SR-SIM), cells were grown on 18 mm precision glass coverslips (Marienfeld Cat# 0117580) that had been precleaned in 1 M HCl for 30 min, washed twice in ddH$_2$O and sterilised in 100% ethanol. Cells were fixed in 3% v/v formaldehyde in PBS, permeabilised with 0.1% saponin in PBS and blocked in 1% BSA/0.01% saponin in PBS. Primary antibody (diluted in BSA block) was added for 1 h at room temperature. Coverslips were washed three times in BSA block and then fluorochrome-conjugated secondary antibody was added in block for 30 min at room temperature. Coverslips were then washed three times in PBS and stained with 300 nM DAPI in PBS for 5 min, before a wash in PBS and a final wash in ddH$_2$O. Apart from iPSC-derived cortical neurons, coverslips were mounted in SlowFade™ Glass Soft-set Antifade Mountant (Thermo Fisher Scientific Cat# S36917) and sealed with CoverGrip™ Coverslip Sealant (Biotium Cat# 23005). For iPSC-derived cortical neurons, coverslips were mounted in ProLong™ Diamond Antifade Mountant (Thermo Fisher Scientific Cat# P36965) and sealed with clear nail polish. For SR-SIM, glass slides were precleaned as described above for the glass coverslips. Channel alignment slides were prepared alongside the SR-SIM samples using multi-coloured fluorescent TetraSpeck™ 0.1 μm Microspheres (Thermo Fisher Scientific Cat# T7279).

**Widefield imaging**. Widefield images were captured on a Leica DMi8 inverted microscope equipped with an iTK LMT200 motorised stage, a 63x/1.47 oil objective (HC PL APO 63x/1.47 OIL) and a Leica DFC9000 GTC Camera, and controlled with LAS X (Leica Application Software X) version 3.5.5.19976. For DAGLB in RUSC2 overexpressing cells, images are representative of two technical replicates of separate coverslips from the same batch of cells, including three independent rescue transfections, with immunofluorescence and microscopy performed independently for each replicate. At least ten images were captured per condition, with both cell selection and manual focus performed on the RUSC2-GFP channel, without viewing the DAGLB channel. Consistent results were observed for cells expressing RUSC2 with an N-terminal GFP tag and with AP4E1 knockout. For HA-DAGLA and HA-DAGLB localisation experiments, images are representative of three independent transfections, with microscopy performed independently for each. For quantification of DAGLB at the TGN in HeLa and SH-SY5Y (undifferentiated and differentiated), experiments were performed in biological triplicate, from separate dishes of cells, and with immunofluorescence and microscopy performed independently for each replicate. Cells were selected for imaging using the DAPI channel only in the Navigator software module of LAS X. For HeLa, 30 images were captured per cell line per replicate, using autofocus on the DAGLB channel. For undifferentiated SH-SY5Y, 20 images were captured per cell line per replicate, with manual focus on the DAGLB channel. For differentiated SH-SY5Y, a minimum of ten images were captured per cell line per replicate, with manual focus on the DAGLB channel. For quantification of DAGLB at the TGN in iPSC-derived cortical neurons, the experiment was performed in technical triplicate with three separate coverslips per cell line from the same batch of cells and with microscopy conducted in three separate sessions. Neurons were selected for imaging using the TUJ1 signal to distinguish neuron cell bodies from astrocytes, without viewing the DAGLB channel. A minimum of 24 images were captured per cell line per replicate, with manual focus on the TGN46 channel.

**Structured illumination microscopy**. Super-resolution structured illumination Microscopy (SR-SIM) was performed on a Zeiss Elyra PS.1 microscope equipped with a 63x/1.46 oil objective (alpha Plan-Apochromat 63x/1.46 Oil Korr M27) and a PCO pco.edge 4.2 sCMOS Camera, and controlled with Zeiss ZEN 2 software with the SR-SIM module. Imaging was performed sequentially, with SERINC1 or SERINC3 (labelled with Alexa Fluor Plus 488) imaged first and DAGLB or ATG9A

(labelled with Alexa Fluor 568) imaged second, using 3 rotations of the grid pattern and 5 phases for each rotation. Experiments were performed in biological duplicate, from separate dishes of cells, and with immunofluorescence and microscopy performed independently for each replicate. 20 cells were imaged per condition per replicate, and cell selection and manual focus were performed on the first acquired channel only, without viewing the second channel. For channel alignment, slides with multi-coloured fluorescent beads were imaged before and after each experiment using the same acquisition settings. The data were processed using the SIM module in Zen 2 software in manual mode, using default settings and a theoretical PSF model, except for the following settings: Noise filter -1; SR frequency weighting 2; Max.Isotrop on; Baseline shifted on; Raw scale on. Channel alignment fit parameters were calculated in Zen 2 software using a multi-coloured bead image as input, in affine mode. The resulting parameters were tested on further bead images and then applied to the experimental images. The channel-aligned images were used for colocalisation analysis as described below.

**Image enhancements.** For representative images of microscopy or Western blot data displayed in figures, global linear brightness and contrast changes were performed in ImageJ[52] version 2.1.0 to enhance visualisation. However, all quantification of such data was performed on the raw unaltered images.

**Colocalisation analysis.** Colocalisation analysis was performed by comparing the fluorescent spots of the two proteins from super-resolution SIM images, following processing and channel alignment as described above. Colocalisation of proteins was quantified in ImageJ[52] version 2.1.0 using the Coloc2 plugin, included in the FIJI package distribution[53]. Individual cells were cropped and manually outlined in ImageJ using the SERINC1 or SERINC3 channel. After this step, the complete analysis pipeline was performed by means of a macro script, available at https://zenodo.org/record/5710246. The functions used at each processing step to identify the spots in the two channels are reported in the following, as described in ImageJ, with parameters in pixels. The pixel size of the images is 32 nm.

Each channel was pre-processed separately by first reducing the image noise (Gaussian Blur, sigma: 1) and then removing the background signal (Subtract Background, rolling ball radius: 20). The spots were identified as regions with intensity twice the standard deviation above the local average signal (Remove Outliers, block radius: 40, standard deviations: 2) and used to generate an initial mask. Small spots were discarded (Analyze Particles, size larger than 4) and the edges of the remaining spots were smoothed (Gaussian Blur, sigma: 0.75). The resulting soft mask was applied to the original intensity image and finally the resulting images from each channel were provided as input to Coloc2. The colocalisation output metric used for analyses was Pearson's Correlation Coefficient. Data were analysed using a two-tailed Mann–Whitney U-test and replicate data were combined for statistical analysis after confirming no significant difference between replicates of the same condition.

**Quantification of DAGLB at the trans-Golgi network.** The accumulation of DAGLB at the TGN was quantified by comparing the fluorescence average intensity in that region with the intensity in the rest of the cell, from widefield images. The analysis was performed with a workflow implemented in the software CellProfiler[54], version 3.1.9, slightly adapted to each cell type analysed. The pipeline files for each cell type are available at https://zenodo.org/record/5710246.

The position and extension of each cell were determined by first identifying the nuclei, using the minimum cross entropy thresholding method on the DAPI channel. The area of each cell was then determined from the DAGLB channel, previously filtered to achieve a uniform distribution of intensity over the cell (two steps of the Smooth operation, first by average filtering, followed by median filtering). Using the nuclei detected as seeds, the area of the cells was obtained from the pre-processed image by watershed combined with Otsu thresholding. The area occupied in each cell by the TGN was determined from the TGN46 channel, after background subtraction, by minimum cross entropy thresholding. This information was used to determine the two complementary regions of each cell where to measure the average fluorescence intensity of the DAGLB signal, after illumination correction. From these measurements a ratio of DAGLB intensity between the TGN and the rest of the cell was calculated. For HeLa and iPSC-derived neurons, ratio values were calculated for each individual cell. For SH-SY5Y, ratio values were not calculated for individual cells, because they were too overlapping. Instead, ratios were calculated for each image, based on the total cell area and total area occupied by the TGN across the entire field of view. In all cases, edge cells were excluded from the analysis.

The workflow was partially modified for the analysis of iPSC-derived cortical neurons, in order to exploit the additional fluorescence signal from the neuron cell body marker TUJ1. Specifically, the marker was used to discriminate neurons from astrocytes, also present in the field of view, and to determine the area of the cells of interest.

Data were analysed using a Kruskal–Wallis test with Dunn's multiple comparisons test for comparisons of more than two conditions or a two-tailed Mann–Whitney U-test when only two conditions were compared. Replicate data were combined for plotting and statistical analysis if there were no significant differences between replicates of the same condition.

**High-throughput confocal imaging.** High-throughput confocal imaging was performed on the ImageXpress Micro Confocal Screening System (Molecular Devices) using the experimental pipeline described by Behne et al.[21] iPSC-derived cortical neurons were grown onto 96-well plates (Greiner Bio-One Cat# 655090) at a density of $20 \times 10^3$ per well and co-cultured with $3 \times 10^3$ iPSC-derived human astrocytes (Astro.4 U, Ncardia). 60 wells per plate were seeded with cells and PBS-only wells included to prevent edge effects. On day 14, plates were fixed using 4% formaldehyde and stained as described above. Images were acquired using a 40X S Plan Fluor objective (NA 0.6, WD 3.6-2.8 mm). Per well, 36 fields were acquired in a 6×6 format. The experiment was performed in biological triplicate, from separate differentiations, and with immunofluorescence and microscopy performed independently for each replicate. Analysis was performed using a customised image analysis pipeline in MetaXpress version 6.7.0.211 (Molecular Devices); settings are available at https://zenodo.org/record/5710246. For the quantification of DAGLB fluorescence shown in Fig. 6c and Supplementary Fig. 5b, neurons were identified based on the presence of DAPI, GOLGA1 and TUJ1 staining, and DAGLB fluorescence intensity was measured in the juxtanuclear region and remaining cell body in each cell. DAGLB fluorescent signal was segmented into a high-intensity area (which overlapped with GOLGA1) and a low-intensity area, based on an arbitrarily defined threshold that was applied uniformly across all experiments. A ratio between the size of these two areas was then calculated on a per cell basis. Cells in which a high-intensity area of DAGLB signal was not identified were excluded from the analysis (this excluded the same proportion of cells from patient and control lines). For each replicate, data were analysed using a two-tailed Mann–Whitney U-test. For quantification of DAGLB puncta in axons, shown in Fig. 6f, neurons were stained and imaged as described above. Per well, 81 fields were acquired in a 9×9 format. Analysis was performed using a customised image analysis pipeline. First, axons were identified based on SMI312 staining and an axonal mask was generated. Next, DAGLB signal was segmented into puncta based on a fluorescence threshold that was uniformly applied across experiments. The number of DAGLB puncta colocalizing with SMI312 positive axons was counted and normalised to the total area occupied by SMI312 staining. Groups were compared using a two-tailed unpaired t test.

**Neurite outgrowth assays.** Automated, non-invasive imaging and measurement of neurite outgrowth in iPSC-derived neurons was performed using the IncuCyte® S3 Live-Cell Analysis System (Sartorius). Day 6 neurons were replated onto 96-well plates (Greiner Bio-One, Cat# 655090) at a density of $10 \times 10^3$ per well. 60 wells per plate were seeded with cells and the remaining wells were filled with PBS to prevent an edge effect. ABX-1431 or DMSO as vehicle control was added to the media to final concentrations indicated in Fig. 7e–h, Supplementary Fig. 6b, d and Supplementary Fig. 7, using the HP D300e digital dispenser (Hewlett Packard). Four hours after replating (t0), plates were moved to the IncuCyte® incubator, and 9 phase-contrast images in a 3 × 3 format were acquired every 3 h for a total of 21 h (until 25 h post-plating) using a 20X S Plan Fluor objective (NA 0.45, WD 8.2–6.9 mm). Neurite length and neurite branch points were quantified using the Incucyte® Neurotrack Analysis Software Module in IncuCyte Controller Version 2020B (Sartorius, Cat# 9600-0010), which performs automated image segmentation of neurites and cell body clusters; settings are available at https://zenodo.org/record/5710246. Processing definitions were optimised for day 6 iPSC-derived cortical neurons and applied to all images analysed. For each 96-well plate the maximum number of images (99) was used to train the segmentation algorithm. Neurite length and number of neurite branch points per image were normalised to the area occupied by cell body clusters (cell body cluster area), followed by normalisation to respective values at the first time point of image acquisition (4 h after replating = t0). Data were analysed using a two-way repeated measures ANOVA with either Šídák's multiple comparisons test or Dunnett's multiple comparisons test, as indicated in the relevant figure legend.

**Lipid extraction.** Brains used for lipid analyses were removed from animals and placed immediately into ice-cold PBS, before being snap frozen in liquid nitrogen. Frozen brain samples were then stored at −80 °C, until the time of lipid extraction. Lipids were extracted with a generic methanol-chloroform-based protocol. Brains from five wild-type (WT) and five *Ap4e1* knockout (KO) mice were weighed and quickly transferred (within seconds and without thawing) to a homogeniser containing 16 mL of cold 2:1:1 (v/v/v) Chloroform:Methanol:Water doped with the following internal standards (all purchased from Cayman Chemical, USA): 2-Arachidonoyl-Glycerol-d8 (2-AG-d8), 1-Stearoyl-2-Arachidonoyl-sn-Glycerol-d8 (SAG-d8), Arachidonic Acid-d8 (AA-d8). Brain homogenates were then vortexed and centrifuged at 850 × *g*, at 4 °C, for 5 min in order to separate the organic fraction. The bottom phase (organic) was collected and the remaining extract was re-extracted with 4 mL $CHCl_3$ containing 40 μL formic acid and centrifuged as described above. The bottom phase (organic) was collected again and both organic phases were combined, vortexed and 250 μL were dried. Finally, dried extracts were resuspended in 250 μL 1-butanol:iso-propanol:$H_2O$ (8:23:69) and directly measured.

**Lipid measurements by nanoLC-timsTOF Pro mass spectrometry.** For measurements of lipids by liquid chromatography coupled to mass spectrometry (LC-

MS), a setup of nanoLC chromatography coupled with ion mobility mass spectrometry was used, as previously described[55]. Briefly, an Easy-nLC 1200 (Thermo Fisher Scientific) ultra-high pressure nanoflow chromatography system was used to separate lipids on an in-house reversed-phase column. The column compartment was heated to 60 °C and lipids were separated with a binary gradient of 35 min at a constant flow rate of 400 nL min$^{-1}$ where mobile phases A and B were acetonitrile:$H_2O$ 60:40% (v/v) and iso-propanol:acetonitrile 90:10% (v/v), both buffered with 0.1% formic acid and 10 mM ammonium formate.

The nanoLC was coupled to a hybrid trapped ion mobility-quadrupole time-of-flight mass spectrometer (timsTOF Pro, Bruker Daltonics, Bremen, Germany) via a modified nanoelectrospray ion source (Captive Spray, Bruker Daltonics). Electrosprayed ions enter the first vacuum stage where they are deflected by 90° and accumulated in the front part of a dual TIMS analyser. Data were acquired with Otofcontrol version 6 (Bruker Daltonics). The instrument was operated with the PASEF method activated in both positive and negative ionisation modes. Each sample was injected in triplicate (injection volume 1 µL in positive and 2 µL in negative mode), in a randomised order, while blanks were scheduled after the triplicate injection of each sample.

**Lipid data analysis**. Processing and analysis of the raw files were performed with MetaboScape 2021 version (Bruker Daltonics, Germany), which contains a feature finding algorithm (T-ReX 4D). This automatically extracts data from the four-dimensional space ($m/z$, retention time, ion mobility and intensity) and assigns MS/MS spectra to them. Precursor ion masses were recalibrated with the lock masses $m/z$ 622.028960 (positive mode) and $m/z$ 666.019887 (negative mode). The raw values of lipid species of 2-AG, SAG and AA were normalised based on the intensity of the corresponding internal standard and the wet tissue weight. Finally, mean values from the triplicate injections were used for statistical analysis. Data were analysed using a two-tailed Mann–Whitney $U$-test.

**Sensitive statistical analysis of AP-4 knockout Dynamic Organellar Maps**. We previously generated six organellar maps from HeLa[2] (duplicates of wild-type, *AP4B1* knockout, and *AP4E1* knockout), and analysed them with our established MR (movement and reproducibility) test[56] to identify proteins with altered subcellular localisation. Briefly, for each protein, the abundance profiles from wild-type and knockout maps were subtracted pairwise, to obtain four sets of 'delta profiles' (wild-type-AP4B1, wild-type-AP4E1, both in duplicate). Proteins that did not shift localisation had delta profiles close to zero. Proteins with significant outlier delta profiles in all replicates (high M(ovement) score, calculated as a robust Mahalanobis Distance within Perseus[57]) were candidate shifting proteins. As a further filter, the similarity of the direction of the shift across replicates was evaluated by calculating delta profile correlation (R(eproducibility) score). To define useful cut offs for M and R, the same analysis was performed on six wild-type HeLa maps[19]. Here, no genuine shifts were expected, so the number of hits identified at a given set of MR cut offs allowed an estimate of the false discovery rate (FDR) for hits detected in the AP-4 knockout vs wild-type analysis. In this previously published study, we used extremely stringent MR settings to filter the data: from the four M scores and the two R scores, we chose the respective lowest values (i.e. most conservative estimate of protein movement). The estimated FDR was <1%, and indeed, all three identified hits were subsequently validated by imaging[2]. Here, we reanalysed this dataset with less stringent filters. We calculated the median M score from the four replicates, and the average of the two R scores. The same calculations were then performed for the six wild-type HeLa maps to estimate the FDR. We chose the same M-score cut-off as previously (M > 4), and an R-score cut-off of >0.92 (the highest value that still included all three previously identified hits). As a further filter, we required the delta profiles of hits to have a correlation >0.9 across both AP-4 knockout clones; this filter was not applied to the wild-type-only dataset. In addition to the previously identified three proteins, we thus identified eight new hits, with an estimated FDR of 25%.

**Shift profile correlation analysis**. To evaluate the similarity of shifts among hits obtained with the organellar maps analysis, we concatenated the delta profiles from all four map comparisons, to obtain one 20-datapoint 'shift' profile for each protein. We then calculated pairwise Pearson's Correlation Coefficients between shift profiles of hits. SERINC1, SERINC3 and ATG9A had previously been identified as AP-4 cargoes, and thus provided reference shift profiles for the identification of new AP-4 cargoes.

**Statistical analysis of TEPSIN-GFP immunoprecipitations**. Sensitive TESPIN-GFP immunoprecipitation SILAC MS data were acquired in our previous study[2]. The processed and filtered dataset described was reanalysed in Perseus[58] version 1.6. A one-sample $t$ test (two-tailed) was applied to compare the log$_2$ L/H ratios for each protein to zero (null hypothesis of no change between TEPSIN-GFP immunoprecipitations and mock immunoprecipitations from wild-type HeLa cells).

**Statistics and reproducibility**. Statistical analyses were performed in GraphPad Prism version 9.1.2 for Windows. No statistical method was used to predetermine sample size. For the SR-SIM colocalisation analyses shown in Fig. 5b, d and Supplementary Fig. 4b, d, sample sizes were chosen based on our previously

published colocalisation analyses of SERINCs and ATG9A[2]. For the high-throughput analyses of DAGLB localisation (Fig. 6c and Supplementary Fig. 5b) and the automated live cell analyses of neurite outgrowth (Fig. 7f–h, Supplementary Fig. 6b–d and Supplementary Fig. 7), sample sizes were chosen based on our previously published high-throughput analyses of ATG9A localisation and analyses of neurite outgrowth in AP-4-deficient iPSC neurons[21]. Generally, quantitative image data underwent threshold filtering prior to analysis, as described in detail above, and edge cells were excluded, with the aim to avoid unreliable or partial measurements near the detection limit of the method. Importantly, no data were ever excluded from the analyses as outliers. The experiments were not randomised. The investigators were not blinded to allocation during experiments and outcome assessment. However, all image analyses were automated to avoid investigator bias.

**Reporting summary**. Further information on research design is available in the Nature Research Reporting Summary linked to this article.

## Data availability

The raw mass spectrometry proteomics data associated with Fig. 1 and Supplementary Fig. 1 were previously published[2] and are available from the ProteomeXchange Consortium via the PRIDE partner repository[59] with the dataset identifier PXD010103. The raw mass spectrometry lipidomics data generated in this study (Fig. 7b–d) have been deposited in the MASS Spectrometry Interactive Virtual Environment (MassIVE) and are accessible via the identifier MSV000088020. The imaging data generated in this study have been deposited on Zenodo and are accessible via the identifiers 5696988[60], 5698395[61] and 5644233[62]. The RNA expression data shown in Supplementary Fig. 3 is publicly available from the Human Protein Atlas[23] via the following links: DAGLB, DAGLA, AP4B1. Source data are provided with this paper.

## Code availability

The image analysis scripts used in this study are available at https://zenodo.org/record/571024663.

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

## Acknowledgements

The authors thank Margaret (Scottie) Robinson for kindly providing the Ap4e1 KO mice, her generous sharing of resources and helpful discussions. We thank Matthias Mann for his continued generous support of the project. Thanks to Jennifer Hirst for sharing of reagents, Paul Manna for advice regarding mouse colony management, Patricia Skow-ronek for help with lipidomics, Julia Schessner and Sebastian Schuck for critical reading of the manuscript and members of the Robinson and Mann Labs for valuable feedback. We are grateful for the dedicated work of the Level 2 technician team at University of Cambridge Central Biomedical Services. Special thanks to the MPIB Imaging Facility for outstanding technical support, in particular to Giovanni Cardone for his advice and assistance with the implementation of image analysis pipelines, as well as feedback on the manuscript, and to Martin Spitaler for his expert technical advice for imaging experiments. This work was funded by the German Research Foundation (DFG/Gottfried Wilhelm Leibniz Prize MA 1764/2-1) and the Max Planck Society for the Advancement of Science. A.K.D. received funding from the European Union's Horizon 2020 research and innovation programme under the Marie Sklodowska-Curie grant agreement no. 896725 and a Humboldt Research Fellowship from the Alexander von Humboldt Foundation. D.E.-F. had support from the CureAP4 Foundation, the Spastic Paraplegia Foundation, the National Institute of Health/National Institute of Neurological Disorders and Stroke (2R25NS070682 and 1K08NS123552-01) and the National Institutes of

Health (BCH IDDRC, NIH P50HD105351). M.Z. received scholarships from the DAAD (German National Exchange Service) and the German National Academic Foundation and is supported by the MD/PhD-Program at Heidelberg University. J.E.A. is supported by the Deutsche Forschungsgemeinschaft (DFG, German Research Foundation) 270949263/GRK2162, the DAAD (German National Exchange Service), the German National Academic Foundation and the Max Weber-Program of the State of Bavaria.

## Author contributions

G.H.H.B. and A.K.D. conceptualised and designed the experiments. D.E.-F. and M.S. designed experiments in iPSC-derived neurons. A.K.D. performed the mouse work, the majority of the microscopy experiments and statistical analyses. G.H.H.B. performed statistical analysis of the Dynamic Organellar Maps data. C.G.V. and F.M. performed the lipidomics analyses. J.E.A., M.Z., H.J., W.A.-S., and D.E.-F. performed the iPSC experiments and high-throughput imaging. A.K.D. and G.H.H.B. wrote the original and revised manuscripts, with contributions from the other authors. G.H.H.B. and D.E.-F. supervised the project.

## Funding

## Competing interests

The authors declare no competing interests.
