## [Peer Review File · Nature Communications]

AP-4-mediated axonal transport controls endocannabinoid production in neuronsREVIEWER COMMENTS

Reviewer #1 (Remarks to the Author):

The present report by Davies and her colleagues stems from their earlier work on AP-4, including the refined reprocessing of proteomic data, and the use of light microscopy and genetics to test if a link exists between AP-4 axonal transport and DAGLB. The hypothesis is interesting. Some of the methods are a bit off in terms of data and image quality. I certainly think that with a substantial revision this paper could significantly be improved. Moreover, there is a number of testable hypotheses on DAGLA and MAGL that could either reinforce or negate the concept presented at the end.

Specific queries:

1. DAGLA is the major DAGL isoform. DAGLB is development-specific. What happens to DAGLA in the samples? Are HeLa and SH-SY5Y cells appropriate to draw conclusions? Do those cells express the complete endocannabinoid cassette? the authors should label DAGLA as well (Fig 1) to reinforce the specificity of a DAGLB-AP-4 interaction.
2. There is a reorganization of DAGL distribution from axonal targeting towards dendritic targeting once a neuron is mature. What happens to AP-4 transport then? If this mechanism is necessary and sufficient to explain DAGLB targeting then this shall as well be reorganized in maturation (or its existence cease entirely). This should be tested in primary neurons (I find it a significant shortcoming that none of the in vitro experiments are on primary cells).
3. A major experimental shortcoming is the lack of ultrastructural data. The super resolution data are just not resolved enough. Electron microscopy needs to be performed throughout and the imaging significantly brushed up. Keimpema et al in 2010 showed that CB1Rs are cargoed on small axonal vesicles. Shall those be distinct from the AP-4 vesicular transport? This should at least be mentioned. Likewise, a detailed search on MAGL, NAPE-PLD, PLCs, CB1Rs, GPR55 should be conducted to increase robustness of a DAGLB-selective mechanism. This is particularly important since CB1R, DAGLs, MAGL, GPR55 co-distribute to axonal growth cones during neurone outgrowth and pathfinding.
4. An estimated FDR of < 25% is unusually high and a relaxed way of looking at proteins. Can the authors use any of the other targets picked up this way to show that DAGLB is not a false-positive hit (i.e. show that other targets are colocalized and cargoed the same way). PTPN9 might be worth following up given that PTPNs (i.e. 22) were previously implicated in endocannabinoid metabolism.
5. "These data suggest that AP-4 deficiency causes a partial block of DAGLB export from the TGN." This statement suggests that alternative mechanisms could exist. What are those? How much of the DAGLB would be recovered at growth cones, and, most notably, how much of the DAGLB enzymatic activity would be retained for cell-autonomous growth? One key experiments in these cell lines would also be the localization of cannabinoid receptors. One might argue that DAGLB products alternative to 2-AG might be missing in the degeneration (AP-4 deficiency) phenotypes that have nothing to do with CB1R-mediated signalling. Therefore, receptor localization (also in HeLa and SH-SY5Y cells) will be needed.
6. "RNA-based expression analysis suggests that DAGLB ..." I think this is a misleading statement. As far as I am aware, the Doherty group specifically placed DAGLB as the early, embryonic isoform of the DAGLs.
7. If AP-4 deficiency increases DAGLB levels then one assumption could be an increased amount of 2-AG. This is an obvious hypothesis to experimentally link 2-AG levels to a biological function. In this context, the Authors should also look at MAGL expression since this might be adaptive, i.e. and increase in MAGL and its axonal localization could render a DAGLB-dependent neurodegeneration hypothesis unfounded. The MAGL aspect is particularly significant when looking

at the AP-4 KO mouse data on 2-AG levels, which could alternatively be explained by an increase in MAGL activity instead of DAGLB inactivation (see again Keimpema et al., 2010). These notions resonate on the conclusions drawn, and an obvious question to address experimentally. Likewise, if DAGLA is the major adult DAGL isoform then its activity might be down-regulated, too. This can be tested in-gel by a method developed by the Cravatt lab.

More minor:
What is TGN?

A phenocopy of the CB1RKO phenotype is quite weak as a supportive argument, and please note that it was the Berghuis et al. 2007 and Mulder et al., 2008 papers describing CB1R genetics and not the Watson et al 2008 paper. It is also noteworthy that DAGLs reorganise in CB1R KO mice, which is not surprising given the axonal phenotypes shown.

Reviewer #2 (Remarks to the Author):

I was invited as technical reviewer regarding lipid analysis for the article by Davies et al.. Generally, the used method is technically sound and valid. The use of timsTOF-MS in conjunction with nanoLC is relatively new but is gaining more and more interest. The use of nanoLC might not have been necessary, but it might have been in place and ready to use and there are no disadvantages.

The only major point I have is, why no full lipidomics analysis was performed on the data? According to the description, the instrument was operated in PASEF mode, which collects full scan and fragmentation data. This could have enabled to compare the two groups based on the full lipid profile and not only selected targets and give the possibility to identify other (important) changes related to the biological question.

Reviewer #3 (Remarks to the Author):

The adaptor protein complex, AP-4, has been reported to play important roles for the intracellular transport of various proteins and the deficiency of AP-4 induces the human disease called AP-4 deficiency syndrome. Therefore, finding the cargo of AP-4 and studying its role in the neuronal function is important for both of biology and medical science. In this manuscript, the authors identified DAGLB as a cargo protein of AP-4 by using spatial proteomics. Then they examine the intracellular distribution in the AP-4 deficient HeLa and SH-SY5Y cells. They also examined the AP-4 deficient effect on DAGLB transport by using IPS cells from the AP-4 deficiency syndrome.

Their data indicates that the DAGLB is one of the cargo proteins of AP-4. However, their neuronal analysis is too immature to conclude that the AP-4 dependent transport of DAGLB really affect the neuronal function.

1. Although the authors indicated that the more amount of DAGLB accumulated in TGN in AP-4 deficient neurons than in the wild type neurons, they don't show the amount of axonal DAGLB is really reduced or not. They need to examine whether the amount of DAGLB in the axon terminal is

reduced in the AP-4 deficient neurons.

2. They indicated that the 2-AG levels are reduced in the brains of AP-4 deficient mice. However, there is no data indicating that the mis-transport of DAGLB directly contribute to the 2-AG level. It is possible that the mis transport of other cargo proteins contributes to the level of 2-AG in brains.

3. They proposed the model in which the mis-transport of DAGLB affects the 2-AG production and activity of CB1/2s reduced. However, they don't show any data suggesting the activities of CB1/2 is affected by the AP-4 deficiency.

Overall, no direct causal relationship between the transport of DAGLB and neuronal function is shown.

I also have specific comments on their data.

1. The immune staining pattern of DAGLB and TGN46, in Figure 2a and c, is not clearly overlapped. Especially, the TGN46 staining in SH-SY5Y (Figure 2c) is too weak to recognize the distribution of TGN.

2. The effect of gRNA on the amount of DAGLB protein is not obvious. They need to quantify the intensities of DAGLB band and analyze whether the gRNA significantly increased the amount of DAGLB.

Point by point response to the reviewers' comments

Preface and key new rescue experiment

We would like to thank all three reviewers for their very insightful comments. Following their suggestions, we have substantially augmented and revised the study.

The reviewers' main concerns were that the original manuscript did not provide direct evidence that our model of AP-4/DAGLB/2-AG function is relevant to neuronal biology, and that we could not exclude that other pathways may contribute to the phenotype. We have now performed several new experiments that address these questions, as detailed below. However, one of these experiments is so broadly informative and supportive of our model, that we would like to describe it up-front, before going into the detailed point-by-point response.

Our original model predicted that in AP-4 deficient neurons, lack of axonal DAGLB causes reduced levels of 2-AG, leading to defects in axon growth, as seen in AP-4 deficient patients. We further proposed that pharmacological modulation of 2-AG metabolism may provide a therapeutic avenue. We have now tested this hypothesis experimentally (new Fig. 7e-h, S6b-d & S7). Using an automated live-cell culture imaging platform, we quantified neurite growth of control and AP-4 patient neurons. As expected, AP-4 patient neurons had significantly shorter neurites, with fewer branch points. We then treated cells with a highly specific inhibitor of MGLL (monoglyceride lipase, also known as MAGL), the enzyme that breaks down 2-AG, to increase cellular 2-AG levels. **MGLL inhibition fully restored neurite growth in AP-4 patient neurons, but did not enhance neurite growth in control neurons.**

In combination with our finding of reduced 2-AG in AP-4 knockout brains, **this experiment strongly supports our model of AP-4/DAGLB function in axonal production of 2-AG, and thus the central conclusions of our study.** Since 2-AG is mainly generated by DAGLs, and only DAGLB is a cargo of AP-4 (new Fig. 4), the most straightforward interpretation of our data is that the lack of 2-AG is caused by missorting of DAGLB. Furthermore, our data demonstrate that this pathogenic effect can be rescued *in vitro* with a drug that is already in clinical trials.

Detailed response to individual comments:

Reviewer #1 (Remarks to the Author):

>The present report by Davies and her colleagues stems from their earlier work on AP-4, including the refined reprocessing of proteomic data, and the use of light microscopy and genetics to test if a link exists between AP-4 axonal transport and DAGLB. The hypothesis is interesting. Some of the methods are a bit off in terms of data and image quality. I certainly think that with a substantial revision this paper could significantly be improved. Moreover, there is a number of testable hypotheses on DAGLA and MAGL that could either reinforce or negate the concept presented at the end.

We would like to thank the reviewer for their interest in our study and for making several important suggestions that have enabled us to improve our manuscript.

Specific queries:

1. DAGLA is the major DAGL isoform. DAGLB is development-specific. What happens to DAGLA in the samples? Are HeLa and SH-SY5Y cells appropriate to draw conclusions? Do those cells express the complete endocannabinoid cassette? the authors should label DAGLA as well (Fig 1) to reinforce the specificity of a DAGLB-AP-4 interaction.

The reviewer raises a very important point – there are two DAGLs, DAGLA and DAGLB, and both are principally responsible for the generation of 2-AG. According to deep RNA sequencing data from the Human Protein Atlas (<http://www.proteinatlas.org>), DAGLB is expressed across all tissues, whereas DAGLA has low expression in most tissues, except brain. Ubiquitous expression of DAGLB fits very well with the ubiquitous expression of AP-4. We have added the expression data as a new supplementary figure (Fig. S3). Consistent with these data, our extensive proteomic analyses did not detect expression of DAGLA in HeLa (Davies et al., 2018; Itzhak et al., 2016). To determine if DAGLA is also a cargo of the AP-4 pathway, we expressed HA-tagged DAGLA and DAGLB in HeLa cells. These HA-constructs had previously been generated and validated by the Doherty group (Zhou et al., 2016). The new data are shown in new Fig. 4.

We observed two things:

1. DAGLA and DAGLB had very different subcellular distributions. While DAGLB partially co-localized with the known AP-4 cargo ATG9A, DAGLA did not co-localize. This suggests major differences in the trafficking of DAGLA and DAGLB.
2. We then expressed the DAGLs in HeLa cells that also express GFP-RUSC2, a protein that drives peripheral accumulation of AP-4 vesicles. We saw re-localization of DAGLB to the cell periphery, but not of DAGLA. Peripheral DAGLB co-localized with other AP-4 cargoes, whereas DAGLA did not. From this highly specific assay we conclude that, unlike DAGLB, DAGLA is not a cargo of AP-4 vesicles.

Our study investigates the DAGLB-AP-4 connection in four different cell types: HeLa, undifferentiated SH-SY5Y neuroblastoma, differentiated SH-SY5Y, and human neurons differentiated from iPSCs. In all four cell types, DAGLB localization depends on AP-4. We thus predict that sorting of DAGLB is a ubiquitous function of AP-4. Since DAGLB is expressed in most tissues (see point 6), it can clearly function independently of the brain endocannabinoid pathway – 2-AG is not only a signalling molecule, but also the precursor of arachidonic acid, and hence an important lipid metabolite. Hence, HeLa cells are an appropriate model to investigate fundamental features of this pathway, and are particularly well-suited to proteomic and trafficking studies. The cellular consequences of AP-4 deficiency, however, are likely to be cell type specific; therefore, our phenotypic characterization focused on neurons.

Following the reviewer's suggestions, we have modified the results and discussion section to clarify these points, including the tissue expression data in Fig. S3 and our new DAGL localization data in Fig. 4.

2. There is a reorganization of DAGL distribution from axonal targeting towards dendritic targeting once a neuron is mature. What happens to AP-4 transport then? If this mechanism is necessary and sufficient to explain DAGLB targeting then this shall as well be reorganized in maturation (or its

existence cease entirely). This should be tested in primary neurons (I find it a significant shortcoming that none of the in vitro experiments are on primary cells).

We agree with the reviewer that the developmental regulation of DAGL localization is an interesting question. While it is known that DAGL activity changes from pre- to post-synaptic localization during development, it has not been systematically established how the two DAGLs differ in this. DAGLA has a well-known post-synaptic role in mature neurons, in retrograde synaptic suppression. However, knockout mouse studies show that DAGLB is dispensable for this role (Gao et al., 2010; Tanimura et al., 2010). The DAGLA C-terminal domain mediates interactions with the Homer adaptor proteins, which are thought to be important for its targeting to the post-synapse, but this domain is lacking in DAGLB, suggesting differential regulation of its localization. Due to lack of a specific validated DAGLB antibody that works in mice (Reisenberg et al., 2012), developmental localization studies have largely focused on DAGLA. While we think that the dynamic re-localization of DAGLs during development is a very important question, it is unfortunately outside the scope of this study, which is primarily concerned with the function of AP-4. Nevertheless, we have made substantial additions to our text to include discussion of this topic. On the other hand, we here identify the first trafficking pathway that mediates differential sorting of DAGLA and DAGLB, and this should stimulate and facilitate further investigations.

The reason why this study did not include work with primary AP-4 knockout neurons is that we unfortunately lost our AP-4 knockout mouse colony at the beginning of the pandemic – the mouse work in our animal facility had to be suspended. All we could do before the shut-down was to prepare and freeze mouse brains, which we later used for the lipidomics analysis in Fig. 7b-d. We hence decided to focus on human neurons differentiated from AP-4 patient iPSCs as the model to investigate AP-4 neuronal function. AP-4 patient-derived neurons provide an established model of AP-4 deficiency, and are used by several groups working in this field. The fact that our MGLL inhibition experiment in human neurons agrees very well with the results and predictions from our AP-4 KO mouse brain lipidomics analysis further cross-validates the use of both models.

3. A major experimental shortcoming is the lack of ultrastructural data. The super resolution data are just not resolved enough. Electron microscopy needs to be performed throughout and the imaging significantly brushed up. Keimpema et al in 2010 showed that CB1Rs are cargoed on small axonal vesicles. Shall those be distinct from the AP-4 vesicular transport? This should at least be mentioned. Likewise, a detailed search on MAGL, NAPE-PLD, PLCs, CB1Rs, GPR55 should be conducted to increase robustness of a DAGLB-selective mechanism. This is particularly important since CB1R, DAGLs, MAGL, GPR55 co-distribute to axonal growth cones during neurone outgrowth and pathfinding.

The purpose of the super-resolution imaging in Fig. 5 is to quantify co-localization of AP-4 cargo proteins in transport vesicles. The resolution provided by SIM is sufficient to resolve these vesicles, as can be seen in the figure. With respect to the reviewer, it is not evident to us how electron microscopy would provide further relevant information in this context, and we hence did not include any. Similarly, the confocal and widefield imaging we performed has adequate resolution for the more macroscopic questions we addressed (e.g., colocalization of DAGLB with the TGN or the axon). We did however follow the reviewer's advice and checked our imaging carefully. In some cases (Fig. 2), individual figure panels had suboptimal contrast in the PDF version we submitted. We improved that by adjusting global contrast and brightness settings. Note that this does not affect the quantification, as this was performed on the raw data.

We agree with the reviewer that the axonal delivery of CB1R and other endocannabinoid related proteins is an important and probably very complex question. The primary aim of our study is to understand AP-4 function though, and not to investigate CB1R trafficking. Our current model of the pathway is that AP-4 sorts a very small number of proteins, in essentially all cell types; DAGLB is one of these cargoes. Our proteomic screen was performed in HeLa cells, and we then hypothesized and subsequently validated that DAGLB is an AP-4 cargo in neurons, where AP-4 function is particularly important for cell homeostasis. We currently have no evidence that the other proteins mentioned by the reviewer are cargoes of AP-4. Furthermore, we would like to share some preliminary mouse brain proteomics data with the reviewer (Review Fig. 1) – these show that the expression levels of the proteins mentioned by the reviewer are largely unaffected in AP-4 knockout mouse brain, suggesting that they are not differentially regulated. (Please note that these are pilot data from a separate ongoing study, and will hence not be include in the current manuscript.)

Following the reviewer’s suggestion, we have augmented the discussion to refer to other components of the endocannabinoid signaling pathway, and now cite important publications probing into this question.

Davies & Borner unpublished data (2021)

Figure 1. Proteomic analysis of brains from *Ap4e1* knockout and wild-type mice. Comparison of protein abundance in lysates prepared from *Ap4e1* knockout and wild-type mouse brains, harvested at 3.5 months ($n = 5$), analysed by label-free quantitative mass spectrometry. >7300 proteins were quantified and data were analysed with a two-tailed t-test: volcano lines indicate the significance threshold (FDR = 0.2). ATG9A level was significantly increased in the *Ap4e1* knockout brains, as previously shown (De Pace et al., 2018; Ivankovic et al., 2020), as was the level of RUSC2, another AP-4 vesicle protein. DAGLB was slightly increased in *Ap4e1* knockout brains ($p = 0.02$), but this did not reach statistical significance after FDR control. DAGLA and other endocannabinoid system proteins, MGLL (MAGL), CNR1 (cannabinoid receptor 1), and NAPEPLD, did not change significantly, and phospholipase C (PLC) proteins were also unaffected.

4. An estimated FDR of < 25% is unusually high and a relaxed way of looking at proteins. Can the authors use any of the other targets picked up this way to show that DAGLB is not a false-positive hit (i.e. show that other targets are colocalized and cargoed the same way). PTPN9 might be worth following up given that PTPNs (i.e. 22) were previously implicated in endocannabinoid metabolism.

The reviewer is correct in pointing out that an FDR of 25% is high – but we chose this lenient cut-off on purpose to achieve higher sensitivity. As detailed in the results, in our previous study we used an FDR cut-off of <1% and identified three hits; we expected all of these to be true hits, and indeed all three were validated (Davies et al., 2018). In the present study, we deliberately set the FDR to 25%; this allowed us to identify 8 additional candidate hits in our initial screen. We expected that several of these were false positives, but some should be true positives. To help decide which ones we should pursue further, we applied several other filters, which allowed us to pinpoint DAGLB as the most likely novel AP-4 cargo protein. DAGLB not only profiles closely with the three previously established AP-4 cargoes, but it is also co-precipitated with AP-4, making it a strong candidate AP-4 vesicle protein (Fig. 1a, b, e). We then used extensive widefield and super-resolution microscopy (Fig. 2, 3, 4, 5) paired with genetics (knockdown, knockout, and rescue experiments) and a highly specific sorting assay to confirm that DAGLB is indeed a cargo of the AP-4 pathway.

We apologize that this has not been clear from the text – we have augmented the results section to explain the rationale of using a relatively high FDR in the first screen, and clarified the various stringency filters used to single out DAGLB for further analysis.

We agree with the reviewer that some of the other candidate hits look promising, especially the endosomal ones (PTPN9 and LNPEP). To stratify hits further, we have now added a new shift profile correlation analysis (new Fig. 1c, d). This revealed that the AP-4 cargoes SERINC1, SERINC3, ATG9A and DAGLB undergo highly correlated localization changes in AP-4 deficient cells, consistent with their trafficking in AP-4 vesicles. Interestingly, the shift profiles of PTPN9 and LNPEP did not correlate with AP-4 cargoes, but were highly correlated with each other (new Fig. 1c and S1c, d). This suggests that PTPN9 and LNPEP are trafficked together in response to AP-4 depletion, and may thus act on the same pathway that responds to lack of AP-4. We have now added a corresponding statement to the results section. Since this is likely to be a secondary effect, and thus only indirectly connected to AP-4, it will be an exciting target for future studies.

5. "These data suggest that AP-4 deficiency causes a partial block of DAGLB export from the TGN." This statement suggests that alternative mechanisms could exist. What are those? How much of the DAGLB would be recovered at growth cones, and, most notably, how much of the DAGLB enzymatic activity would be retained for cell-autonomous growth? One key experiments in these cell lines would also be the localization of cannabinoid receptors. One might argue that DAGLB products alterantive to 2-AG might be missing in the degeneration (AP-4 deficiency) phenotypes that have nothing to do with CB1R-mediated signalling. Therefore, receptor localization (also in HeLa and SH-SY5Y cells) will be needed.

We observed accumulation of DAGLB at the TGN in all AP-4 deficient cells we investigated. Not all DAGLB is retained at the TGN – there are still considerable vesicular/organelar pools of DAGLB. We thus agree with the reviewer that there must be additional mechanisms for TGN export. As we have shown previously, this is also the case for the other AP-4 cargoes, ATG9A, SERINC1 and SERINC3 (Davies et al., 2018). AP-4 appears to be a rather specialized TGN export pathway, which mediates highly selective sorting and targeting of a small subset of cargo proteins, probably for dedicated biological activities (e.g., autophagosome biogenesis via sorting of ATG9A). However, the known AP-4 cargoes

have additional established functions/localizations in other parts of the cell (for example, at the plasma membrane). Hence, AP-4 mediates only a part of their TGN export, but clearly an important one.

Following the reviewers' suggestion, we have imaged and quantified DAGLB in the axons of control and AP-4 patient iPSC-derived neurons (new Fig. 6e, f). AP-4 deficiency caused a highly significant drop in the density of DAGLB positive vesicles in the axon ($p < 0.001$). We estimate a reduction of 21% - which is similar to the partial reduction of ATG9A vesicles in primary neuron axons from AP-4 deficient mice (Ivankovic et al., 2020). Clearly, alternative or compensatory mechanisms exist to deliver DAGLB and ATG9A to the axon – but judging from the phenotype, these seem to be insufficient to maintain axonal homeostasis. Although this interplay is intriguing, our current study focuses on the functions of the AP-4 pathway, and we suggest that the identification of alternate sorting mechanisms of DAGLB should be a subject for future investigations.

Regarding the localization of the cannabinoid receptor, we agree with the reviewer that this is generally important. As explained above (point 3), a central result of this study is the discovery that AP-4 sorts DAGLB in neurons, and we currently have no evidence that AP-4 sorts any other components of the cannabinoid signalling cassette. AP-4/DAGLB probably function in diverse cell types, including those that do not express the cannabinoid receptor (such as HeLa). Indeed, the reviewer raises the potential concern that the missorting of DAGLB may have effects unrelated to 2-AG-mediated cannabinoid receptor signalling. While we think it is unlikely that localizing the CB1 receptor would be informative in this case, we agree that ruling out effects unrelated to 2-AG metabolism is important. Therefore, we performed the extensive MGLL inhibitor experiment detailed above. This shows that 2-AG levels become limiting for neurite growth in AP-4 deficient neurons. Raising 2-AG levels fully rescues the phenotype (new Fig. 7e-h, S6b-d and S7). Furthermore, since we also show that only DAGLB is a cargo of AP-4, but not DAGLA (new Fig. 4), this strongly argues that a key defect in AP-4 deficiency is lower 2-AG caused by lack of axonal DAGLB. Since 2-AG drives axonal growth by activating the cannabinoid receptor, our new model (Fig. 8) provides the simplest explanation for axonal growth defects in AP-4 deficiency.

6. "RNA-based expression analysis suggests that DAGLB ..." I think this is a misleading statement. As far as I am aware, the Doherty group specifically placed DAGLB as the early, embryonic isoform of the DAGLs.

As discussed above in point 1, deep RNA sequencing data from the Human Protein Atlas show that DAGLB is ubiquitously expressed in human tissues, while DAGLA is more brain specific. We have now clarified this point by including the actual expression data from the Human Protein Atlas as a new supplemental figure (Fig. S3). We agree with the reviewer that the relative expression levels and patterns of DAGLA and DAGLB during neuronal development are an important question, and we have added this point to the discussion.

7. If AP-4 deficiency increases DAGLB levels then one assumption could be an increased amount of 2-AG. This is an obvious hypothesis to experimentally link 2-AG levels to a biological function. In this context, the Authors should also look at MAGL expression since this might be adaptive, i.e. and increase in MAGL and its axonal localization could render a DAGLB-dependent neurodegeneration hypothesis unfounded. The MAGL aspect is particularly significant when looking at the AP-4 KO mouse data on 2-AG levels, which could alternatively be explained by an increase in MAGL activity instead of DAGLB inactivation (see again Keimpema et al., 2010). These notions resonate on the conclusions drawn, and are an obvious question to address experimentally. Likewise, if DAGLA is the major adult DAGL isoform

then its activity might be down-regulated, too. This can be tested in-gel by a method developed by the Cravatt lab.

The reviewer raises several important points about potential alternative explanations for the observed drop in 2-AG in the brains of AP-4-deficient mice. We have addressed these concerns experimentally and with clarifications to the text, and multiple lines of evidence now support that the drop is indeed caused by reduced DAGLB activity, and not by altered MGLL (MAGL) or DAGLA activity.

First, we have added a simple schematic of the 2-AG pathway to Fig. 7a, to help readers follow the logic of our experiments and conclusions. 2-AG is predominantly generated by two related enzymes, DAGLA and DAGLB (Bisogno et al., 2003). 2-AG is then hydrolyzed by MGLL, to generate arachidonic acid (AA). In principle, lower 2-AG levels could either arise from reduced DAGL activity, or from enhanced MGLL activity.

Importantly, in DAGL deficient mice, 2-AG and AA levels show a correlated drop – thus it appears that DAGLs indirectly control levels of AA (Gao et al., 2010). This observation can help us to understand which enzymatic block causes altered levels of 2-AG: Our AP-4 knockout mouse lipidomics data (Fig. 7b-d) show a parallel decrease in 2-AG and AA, consistent with a drop in DAGL activity. If MGLL activity was increased, we would expect a drop in 2-AG, but no drop in AA. Hence, our lipidomics data indicate that the drop in 2-AG is caused by lack of DAGL activity.

We have also shown conclusively that DAGLB is a cargo of the AP-4 pathway, and that it is mis-trafficked in AP-4 deficient neurons and other cell types. We now provide new evidence that DAGLA is not a cargo of the AP-4 pathway (new Fig. 4). This further suggests that the lower 2-AG levels are caused by mis-trafficking of DAGLB.

Furthermore, we would like to refer again to our preliminary mouse brain proteomics data (Review Fig. 1). Our data show that the expression levels of MGLL and DAGLA are almost identical in control and *Ap4e1* knockout mice. This further suggests that these two enzymes do not contribute to the 2-AG phenotype.

Finally, our MGLL inhibitor rescue experiment in human iPSC neurons is completely consistent with the predictions from the mouse lipidomics data, and cross-corroborates the findings in both systems.

Taken together, the simplest explanation of our data is that the 2-AG phenotype in AP-4 deficient brains is caused by reduced activity of DAGLB due to mis-trafficking. Even the slight upregulation of DAGLB expression can clearly not compensate its lower activity, further supporting that the correct subcellular localization is very important for DAGL function.

More

minor:

What is TGN?

TGN is the commonly used acronym for ‘*trans*-Golgi network’. We had accidentally edited out the introduction of this abbreviation and thank the reviewer for spotting this mistake. We have added the definition to where we first use ‘TGN’ in the main text and have replaced ‘TGN’ in the abstract with ‘*trans*-Golgi network’.

A phenocopy of the CB1RKO phenotype is quite weak as a supportive argument, and please note that it was the Berghuis et al. 2007 and Mulder et al., 2008 papers describing CB1R genetics and not the

Watson et al 2008 paper. It is also noteworthy that DAGLs reorganise in CB1R KO mice, which is not surprising given the axonal phenotypes shown.

We apologize for this inaccuracy – we have added references to these papers accordingly.

Reviewer #2 (Remarks to the Author):

I was invited as technical reviewer regarding lipid analysis for the article by Davies et al.. Generally, the used method is technically sound and valid. The use of timsTOF-MS in conjunction with nanoLC is relatively new but is gaining more and more interest. The use of nanoLC might not have been necessary, but it might have been in place and ready to use and there are no disadvantages. The only major point I have is, why no full lipidomics analysis was performed on the data? According to the description, the instrument was operated in PASEF mode, which collects full scan and fragmentation data. This could have enabled to compare the two groups based on the full lipid profile and not only selected targets and give the possibility to identify other (important) changes related to the biological question.

We would like to thank the reviewer for their positive assessment of our lipidomics data. As the reviewer mentioned, our analysis workflow allowed us to acquire data for lipids beyond the targeted lipids AA, 2-AG and SAG. However, the analysis of such data is not trivial, and requires many manual filtering, identification and quantification steps. Since the present study investigates the role of AP-4 in endocannabinoid metabolism, we thought it best to focus our analysis to the three most relevant lipids. We feel that further lipidomics data without any follow up may potentially add ‘loose ends’ and distract from the central narrative of the manuscript. We are however planning to analyze the complete lipidomics data, and this will hopefully be part of a future investigation. Nevertheless, we are happy to show the complete raw data to the reviewer, as was suggested by the editor. For this purpose, we have uploaded the data onto the Massive repository, from where they can be accessed via the URL: <ftp://MSV000088020@massive.ucsd.edu> (username: vasilopoulou; password: lipidomics). Please feel free to inspect the data.

Reviewer #3 (Remarks to the Author):

The adaptor protein complex, AP-4, has been reported to play important roles for the intracellular transport of various proteins and the deficiency of AP-4 induces the human disease called AP-4 deficiency syndrome. Therefore, finding the cargo of AP-4 and studying its role in the neuronal function is important for both of biology and medical science. In this manuscript, the authors identified DAGLB as a cargo protein of AP-4 by using spatial proteomics. Then they examine the intracellular distribution in the AP-4 deficient HeLa and SH-SY5Y cells. They also examined the AP-4 deficient effect on DAGLB transport by using IPS cells from the AP-4 deficiency syndrome.

We would like to thank the reviewer for supporting the importance of our study.

Their data indicates that the DAGLB is one of the cargo proteins of AP-4. However, their neuronal analysis is too immature to conclude that the AP-4 dependent transport of DAGLB really affect the neuronal function.

Following the reviewer's suggestions, we have now greatly strengthened the neuronal investigation of the AP-4 pathway (details below). We have substantially augmented the imaging section on DAGLB localization in neurons (new Fig. 6e, f and S5b). Most importantly, our new MGLL inhibition experiment (new Fig. 7e-h, S6b-d and S7) provides strong evidence that the AP-4 axonal growth defect is caused by lack of 2-AG/DAGLB missorting.

1. Although the authors indicated that the more amount of DAGLB accumulated in TGN in AP-4 deficient neurons than in the wild type neurons, they don't show the amount of axonal DAGLB is really reduced or not. They need to examine whether the amount of DAGLB in the axon terminal is reduced in the AP-4 deficient neurons.

This is an important point, which we have now addressed experimentally (new Fig. 6e, f). We have quantified the number of DAGLB positive vesicles per μm^2 in the axons of AP-4 patient and control neurons, using unbiased automated high-throughput imaging. Indeed, we observed a highly significant reduction of DAGLB in the axons of AP-4 patient neurons ($p < 0.001$). The level of depletion (21%) is similar to that reported for Atg9a in the axons of AP-4 deficient primary mouse neurons (Ivankovic et al., 2020). These data strongly support that AP-4 targets DAGLB to the axon.

2. They indicated that the 2-AG levels are reduced in the brains of AP-4 deficient mice. However, there is no data indicating that the mis-transport of DAGLB directly contribute to the 2-AG level. It is possible that the mis transport of other cargo proteins contributes to the level of 2-AG in brains.

Several research groups have provided compelling evidence that 2-AG is predominantly generated by two related enzymes, DAGLA and DAGLB (Bisogno et al., 2003; Gao et al., 2010; Tanimura et al., 2010). 2-AG is then hydrolyzed by MGLL, to generate arachidonic acid (AA). In principle, lower 2-AG levels could either arise from reduced DAGL activity, or from enhanced MGLL activity. We have added a simple schematic of the 2-AG pathway to new Fig. 7a, to help readers follow the logic of our experiments.

Importantly, published evidence shows that in DAGL deficient mice, 2-AG and AA levels show a correlated drop – thus it appears that DAGLs indirectly control levels of AA (Gao, et al., 2010). This observation can help us to understand which enzymatic block causes altered levels of 2-AG: Our AP-4 knockout mouse lipidomics data show a concomitant decrease in 2-AG and AA, consistent with a drop in DAGL activity. If MGLL activity was increased, we would not expect a drop in AA. Hence, our lipidomics data indicate that the drop in 2-AG is caused by lack of DAGL activity. We have added several sentences to the results and discussion section to clarify the logic of this experiment and to aid the reader in their interpretation of our results.

We have shown conclusively that DAGLB is a cargo of the AP-4 pathway, and that it is mis-trafficked in AP-4 deficient neurons and other cell types. We now provide new evidence that DAGLA is not a cargo of the AP-4 pathway (new Fig. 4). This further suggests that the lower 2-AG levels are caused by mis-trafficking of DAGLB, and not by other cargo proteins.

Furthermore, we would like to refer the reviewer to our preliminary mouse brain proteomics data (Review Fig. 1). We have quantified protein expression in brains from control and AP-4 knockout mice.

Our data show that the expression levels of MGLL and DAGLA are almost identical in control and Ap4e1 knockout mice. This further suggests that these two enzymes do not contribute to the 2-AG phenotype, and other components of the endocannabinoid signaling network were also unaffected.

Finally, our MGLL rescue experiment in human iPSC neurons (new Fig. 7e-h, S6b-d and S7), detailed above, is completely consistent with the predictions from the mouse lipidomics data, and cross-corroborates the findings in both systems.

Taken together, the simplest explanation of our data is that the 2-AG phenotype in AP-4 deficient neurons is caused by mis-trafficking of DAGLB.

3. They proposed the model in which the mis-transport of DAGLB affects the 2-AG production and activity of CB1/2s reduced. However, they don't show any data suggesting the activities of CB1/2 is affected by the AP-4 deficiency.

In this study we have shown that AP-4 deficiency leads to mistrafficking of DAGLB and reduced brain levels of 2-AG. 2-AG is the principle agonist of the CB1 receptor in brain, and both 2-AG and CB1R have well-established roles in axon growth. Therefore, in the first version of our manuscript we hypothesised that axon growth defects in AP-4 deficiency may arise due to impaired DAGLB activity, as a consequence of its mistrafficking. We have now confirmed this hypothesis with our MGLL inhibition experiment (new Fig. 7e-h, S6b-d and S7), as detailed above. By inhibiting MGLL to raise 2-AG levels in AP-4-deficient neurons, we restored normal neurite growth. Based on the established model of 2-AG/CB1R function in axonal growth, the simplest explanation is that the rescue occurs via restored CB1R signalling.

Overall, no direct causal relationship between the transport of DAGLB and neuronal function is shown.

We hope that we have convinced the reviewer that our MGLL inhibitor experiment provides strong evidence that the neurite outgrowth defect in AP-4 deficient cells is caused by a lack of 2-AG, which is most simply explained by missorting of DAGLB. We currently have no data that contradict this simple model, and it agrees very well with the existing literature.

I also have specific comments on their data.

1. The immune staining pattern of DAGLB and TGN46, in Figure 2a and c, is not clearly overlapped. Especially, the TGN46 staining in SH-SY5Y (Figure 2c) is too weak to recognize the distribution of TGN.

We have carefully reviewed the images in Fig. 2. We agree with the reviewer that some panels had poor brightness/contrast settings and thank them for spotting this mistake. We have now adjusted these (only linear adjustments affecting the whole image were performed), and hope that the images are clearer now. In any case, the difference in DAGLB localization between wild-type and AP-4-deficient cells is rather subtle, especially in HeLa cells, and cannot be judged accurately from single images. That is why we used unbiased automated image analysis to quantify the effect (which is performed on the raw microscopy data).

2. The effect of gRNA on the amount of DAGLB protein is not obvious. They need to quantify the intensities of DAGLB band and analyze whether the gRNA significantly increased the amount of DAGLB.

We have now quantified the amount of DAGLB by densitometry (new Fig. 2f). There is a clear increase in DAGLB in the AP-4-depleted SH-SY5Y cells.

Additional references now cited in this study

- Berghuis, P., Rajnicek, A. M., Morozov, Y. M., Ross, R. A., Mulder, J., Urbán, G. M., Monory, K., Marsicano, G., Matteoli, M., Canty, A., Irving, A. J., Katona, I., Yanagawa, Y., Rakic, P., Lutz, B., Mackie, K., & Harkany, T. (2007). Hardwiring the Brain: Endocannabinoids Shape Neuronal Connectivity. *Science*, 316(5828), 1212 LP – 1216. <https://doi.org/10.1126/science.1137406>
- Cisar, J. S., Weber, O. D., Clapper, J. R., Blankman, J. L., Henry, C. L., Simon, G. M., Alexander, J. P., Jones, T. K., Ezekowitz, R. A. B., O'Neill, G. P., & Grice, C. A. (2018). Identification of ABX-1431, a Selective Inhibitor of Monoacylglycerol Lipase and Clinical Candidate for Treatment of Neurological Disorders. *Journal of Medicinal Chemistry*, 61(20), 9062–9084. <https://doi.org/10.1021/acs.jmedchem.8b00951>
- Claude-Taupin, A., Jia, J., Bhujabal, Z., Garfa-Traoré, M., Kumar, S., da Silva, G. P. D., Javed, R., Gu, Y., Allers, L., Peters, R., Wang, F., da Costa, L. J., Pallikkuth, S., Lidke, K. A., Mauthe, M., Verlhac, P., Uchiyama, Y., Salemi, M., Phinney, B., ... Deretic, V. (2021). ATG9A protects the plasma membrane from programmed and incidental permeabilization. *Nature Cell Biology*. <https://doi.org/10.1038/s41556-021-00706-w>
- D'Amore, A., Tessa, A., Naef, V., Bassi, M. T., Citterio, A., Romaniello, R., Fichi, G., Galatolo, D., Mero, S., Battini, R., Bertocci, G., Baldacci, J., Sicca, F., Gemignani, F., Ricca, I., Rubegni, A., Hirst, J., Marchese, M., Sahin, M., ... Santorelli, F. M. (2020). Loss of ap4s1 in zebrafish leads to neurodevelopmental defects resembling spastic paraplegia 52. *Annals of Clinical and Translational Neurology*, 7(4), 584–589. <https://doi.org/10.1002/acn3.51018>
- Deng, H., & Li, W. (2020). Monoacylglycerol lipase inhibitors: modulators for lipid metabolism in cancer malignancy, neurological and metabolic disorders. *Acta Pharmaceutica Sinica B*, 10(4), 582–602. <https://doi.org/10.1016/j.apsb.2019.10.006>
- Ebrahimi-Fakhari, D., Cheng, C., Dies, K., Diplock, A., Pier, D. B., Ryan, C. S., Lanpher, B. C., Hirst, J., Chung, W. K., Sahin, M., Rosser, E., Darras, B., & Bennett, J. T. (2018). Clinical and genetic characterization of AP4B1-associated SPG47. *American Journal of Medical Genetics. Part A*, 176A, 311–318. <https://doi.org/10.1002/ajmg.a.38561>
- Gao, Y., Vasilyev, D. V., Goncalves, M. B., Howell, F. V., Hobbs, C., Reisenberg, M., Shen, R., Zhang, M.-Y., Strassle, B. W., Lu, P., Mark, L., Piesla, M. J., Deng, K., Kouranova, E. V., Ring, R. H., Whiteside, G. T., Bates, B., Walsh, F. S., Williams, G., ... Doherty, P. (2010). Loss of Retrograde Endocannabinoid Signaling and Reduced Adult Neurogenesis in Diacylglycerol Lipase Knock-out Mice. *Journal of Neuroscience*, 30(6), 2017–2024. <https://doi.org/10.1523/JNEUROSCI.5693-09.2010>
- Guardia, C. M., Jain, A., Mattera, R., Friefeld, A., Li, Y., & Bonifacino, J. S. (2021). RUSC2 and WDR47 oppositely regulate kinesin-1 – dependent distribution of ATG9A to the cell periphery. *Molecular Biology of the Cell*, Aug 25:mbc, Epub ahead of print. <https://doi.org/10.1091/mbc.E21-06-0295>

- Keimpema, E., Barabas, K., Morozov, Y. M., Tortoriello, G., Torii, M., Cameron, G., Yanagawa, Y., Watanabe, M., Mackie, K., & Harkany, T. (2010). Differential subcellular recruitment of monoacylglycerol lipase generates spatial specificity of 2-arachidonoyl glycerol signaling during axonal pathfinding. *Journal of Neuroscience*, *30*(42), 13992–14007. <https://doi.org/10.1523/JNEUROSCI.2126-10.2010>
- Nomura, D. K., Morrison, B. E., Blankman, J. L., Long, J. Z., Kinsey, S. G., Marcondes, M. C. G., Ward, A. M., Hahn, Y. K., Lichtman, A. H., Conti, B., & Cravatt, B. F. (2011). Endocannabinoid Hydrolysis Generates Brain Prostaglandins That Promote Neuroinflammation. *Science*, *334*(November), 809–814.
- Reisenberg, M., Singh, P. K., Williams, G., & Doherty, P. (2012). The diacylglycerol lipases: Structure, regulation and roles in and beyond endocannabinoid signalling. *Philosophical Transactions of the Royal Society B: Biological Sciences*, *367*(1607), 3264–3275. <https://doi.org/10.1098/rstb.2011.0387>
- Schlosburg, J. E., Blankman, J. L., Long, J. Z., Nomura, D. K., Pan, B., Kinsey, S. G., Nguyen, P. T., Ramesh, D., Booker, L., Burston, J. J., Thomas, E. A., Selley, D. E., Sim-Selley, L. J., Liu, Q. S., Lichtman, A. H., & Cravatt, B. F. (2010). Chronic monoacylglycerol lipase blockade causes functional antagonism of the endocannabinoid system. *Nature Neuroscience*, *13*(9), 1113–1119. <https://doi.org/10.1038/nn.2616>
- Tanimura, A., Yamazaki, M., Hashimotodani, Y., Uchigashima, M., Kawata, S., Abe, M., Kita, Y., Hashimoto, K., Shimizu, T., Watanabe, M., Sakimura, K., & Kano, M. (2010). The Endocannabinoid 2-Arachidonoylglycerol Produced by Diacylglycerol Lipase α Mediates Retrograde Suppression of Synaptic Transmission. *Neuron*, *65*(3), 320–327. <https://doi.org/10.1016/j.neuron.2010.01.021>
- Zhou, Y., Howell, F. V., Glebov, O. O., Albrecht, D., Williams, G., & Doherty, P. (2016). Regulated endosomal trafficking of Diacylglycerol lipase alpha (DAGL α) generates distinct cellular pools; implications for endocannabinoid signaling. *Molecular and Cellular Neuroscience*, *76*, 76–86. <https://doi.org/10.1016/j.mcn.2016.08.011>
- Ziegler, M., Russell, B. E., Eberhardt, K., Geisel, G., D'Amore, A., Sahin, M., Kornblum, H. I., & Ebrahimi-Fakhari, D. (2021). Blended Phenotype of Silver-Russell Syndrome and SPG50 Caused by Maternal Isodisomy of Chromosome 7. *Neurology Genetics*, *7*(1), e544. <https://doi.org/10.1212/nxg.0000000000000544>

REVIEWERS' COMMENTS

Reviewer #1 (Remarks to the Author):

The authors performed extensive experimentation during the revision. Their MAGLL rescue experiment is impressive and supportive of the original hypothesis. By and large, they have addressed most of my concerns adequately. Therefore, I support publication of their study.

Reviewer #2 (Remarks to the Author):

My questions regarding the full lipidomic analysis has been answered. I welcome the decision to upload the data to MASSIVE, but I would ask to remove password protection.

Reviewer #3 (Remarks to the Author):

The authors made substantial changes to the manuscripts in response to the reviewers' comments. In particular, they provided the new important evidence by MGLL inhibition experiment. I have no further concern about their experiments and I think their manuscript is now suitable for the publication in Nature Communications.

RESPONSE TO REVIEWERS' COMMENTS

Reviewer #1 (Remarks to the Author): The authors performed extensive experimentation during the revision. Their MAGLL rescue experiment is impressive and supportive of the original hypothesis. By and large, they have addressed most of my concerns adequately. Therefore, I support publication of their study.

We would like to thank the reviewer for their positive evaluation of our revision. No further questions are raised.

Reviewer #2 (Remarks to the Author): My questions regarding the full lipidomic analysis has been answered. I welcome the decision to upload the data to MASSIVE, but I would ask to remove password protection.

We have removed the password protection, as requested by the reviewer.

Reviewer #3 (Remarks to the Author): The authors made substantial changes to the manuscripts in response to the reviewers' comments. In particular, they provided the new important evidence by MGLL inhibition experiment. I have no further concern about their experiments and I think their manuscript is now suitable for the publication in Nature Communications.

We would like to thank the reviewer for their positive evaluation of our revision. No further questions are raised.